# Metabolic shift underlies recovery in reversible infantile respiratory chain deficiency

Denisa Hathazi[1,†], Helen Griffin[2,†], Matthew J Jennings[1,†], Michele Giunta[2,†], Christopher Powell[3] ⓘD,
Sarah F Pearce[3,4], Benjamin Munro[1], Wei Wei[1,3], Veronika Boczonadi[2], Joanna Poulton[5], Angela Pyle[2],
Claudia Calabrese[1,3], Aurora Gomez-Duran[1,3], Ulrike Schara[6], Robert D S Pitceathly[7], Michael G
Hanna[7], Kairit Joost[8] ⓘD, Ana Cotta[9], Julia Filardi Paim[9], Monica Machado Navarro[10], Jennifer Duff[2],
Andre Mattman[11,‡], Kristine Chapman[11], Serenella Servidei[12], Adela Della Marina[6],
Johanna Uusimaa[13], Andreas Roos[6,14], Vamsi Mootha[15], Michio Hirano[16], Mar Tulinius[17], Mamta Giri[18,‡],
Eric P Hoffmann[18], Hanns Lochmüller[19,20,21,22], Salvatore DiMauro[16], Michal Minczuk[3] ⓘD,
Patrick F Chinnery[1,3] ⓘD, Juliane S Müller[1,†] & Rita Horvath[1,*] ⓘD

## Abstract

Reversible infantile respiratory chain deficiency (RIRCD) is a rare mitochondrial myopathy leading to severe metabolic disturbances in infants, which recover spontaneously after 6-months of age. RIRCD is associated with the homoplasmic m.14674T>C mitochondrial DNA mutation; however, only ~ 1/100 carriers develop the disease. We studied 27 affected and 15 unaffected individuals from 19 families and found additional heterozygous mutations in nuclear genes interacting with mt-tRNAGlu including *EARS2* and *TRMU* in the majority of affected individuals, but not in healthy carriers of m.14674T>C, supporting a digenic inheritance. Our transcriptomic and proteomic analysis of patient muscle suggests a stepwise mechanism where first, the integrated stress response associated with increased FGF21 and GDF15 expression enhances the metabolism modulated by serine biosynthesis, one carbon metabolism, TCA lipid oxidation and amino acid availability, while in the second step mTOR activation leads to increased mitochondrial biogenesis. Our data suggest that the spontaneous recovery in infants with digenic mutations may be modulated by the above described changes. Similar mechanisms may explain the variable penetrance and tissue specificity of other mtDNA mutations and highlight the potential role of amino acids in improving mitochondrial disease.

**Keywords** digenic inheritance; homoplasmic tRNA mutation; mitochondrial myopathy; reversible infantile respiratory chain deficiency
**Subject Categories** Membrane & Trafficking; Metabolism
**The EMBO Journal (2020) 39: e105364**

1  Department of Clinical Neurosciences, School of Clinical Medicine, University of Cambridge, Cambridge, UK
2  Wellcome Centre for Mitochondrial Research, Translational and Clinical Research Institute, Newcastle University, Newcastle upon Tyne, UK
3  MRC Mitochondrial Biology Unit, Cambridge, UK
4  Karolinska Institute, University of Stockholm, Stockholm, Sweden
5  Nuffield Department of Obstetrics & Gynaecology, University of Oxford, Oxford, UK
6  Pediatric Neurology, University of Essen, Essen, Germany
7  Department of Neuromuscular Diseases, UCL Queen Square Institute of Neurology and The National Hospital for Neurology and Neurosurgery, London, UK
8  Centre of Allergology and Immunology, East-Tallinn Central Hospital, Tallinn, Estonia
9  Department of Pathology, Neuromuscular Unit, SARAH Network of Rehabilitation Hospitals, Belo Horizonte, Brazil
10 Department of Pediatrics, Neuromuscular Unit, SARAH Network of Rehabilitation Hospitals, Belo Horizonte, Brazil
11 Department of Pathology and Laboratory Medicine SPH, St. Paul's Hospital, Vancouver, BC, Canada
12 Department of Neurology, Università Cattolica del Sacro Cuore, Roma, Italy
13 PEDEGO Research Unit/Pediatric Neurology, Medical Research Center Oulu, Oulu University Hospital and University of Oulu, Oulu, Finland
14 Leibniz Institute for Analytical Sciences (ISAS), Dortmund, Germany
15 Howard Hughes Medical Institute, Harvard Medical School, Boston, MA, USA
16 Department of Neurology, Columbia University Medical Center, New York, NY, USA
17 Department of Pediatrics, The Sahlgrenska Academy, University of Gothenburg, Gothenburg, Sweden
18 School of Pharmacy and Pharmaceutical Sciences, Binghamton University, New York, NY, USA
19 Department of Neuropediatrics and Muscle Disorders, Medical Center, Faculty of Medicine, University of Freiburg, Freiburg, Germany
20 Centro Nacional de Análisis Genómico (CNAG-CRG), Center for Genomic Regulation, Barcelona Institute of Science and Technology (BIST), Barcelona, Spain
21 Children's Hospital of Eastern Ontario Research Institute, University of Ottawa, Ottawa, ON, Canada
22 Division of Neurology, Department of Medicine, The Ottawa Hospital, Ottawa, ON, Canada
   *Corresponding author. Tel: +44 (0) 1223 762092; E-mail: rh732@medschl.cam.ac.uk
   †These authors contributed equally to this work
   ‡Correction added on 1 December 2020, after first online publication: The spelling of the author names Andre Mattman and Mamta Giri was corrected.

# Introduction

Mitochondrial diseases are a large and clinically heterogeneous group of disorders that result from deficiencies in cellular energy production and affect at least one in 4,300 of the population (Gorman *et al*, 2016). Although defective oxidative phosphorylation is the common pathway behind the disease, it is unknown why different mtDNA or nuclear mutations result in largely heterogeneous and often tissue-specific clinical presentations.

We have previously studied a rare and unique group of mitochondrial diseases, where life-threatening symptoms present in infancy, but recover spontaneously after 6 months of age (infantile reversible mitochondrial diseases) (Boczonadi *et al*, 2015). The most common form of these conditions, reversible infantile respiratory chain deficiency (RIRCD OMIM *500009), previously called reversible infantile cytochrome c oxidase deficiency myopathy (Horvath *et al*, 2009) is characterised by severe muscular hypotonia and weakness before 3 months of age (floppy baby), followed by a complete or almost complete spontaneous recovery after 6 months of age. Other organs are usually not involved. All patients affected by RIRCD carry the homoplasmic m.14674T>C/G mt-tRNA$^{Glu}$ mutation, but only about a third of carriers within RIRCD families develop symptoms. Despite the striking reversible phenotype, < 100 RIRCD patients have been described worldwide. The carrier frequency of the m.14674T>C variant is 0.01% in the healthy population (https://www.hmtdb.uniba.it/) and at a much higher frequency, 2/1,000 (0.2%) in Oceania, linked to the presence of this variant on the M27b1 mtDNA haplogroup; however, no data are publicly available on the prevalence of RIRCD in Oceanian population (Duggan *et al*, 2014). The reasons for the markedly reduced penetrance are unknown.

A few other mitochondrial diseases have been also associated with a reversible infantile disease course (Boczonadi *et al*, 2015). Reversible infantile hepatopathy is caused by autosomal recessive mutations in the *TRMU* gene (tRNA 5-methylaminomethyl-2-thiouridylate methyltransferase, OMIM *610230) (Zeharia *et al*, 2009), which uses cysteine to thiouridylate mt-tRNA$^{Glu}$, mt-tRNA$^{Gln}$ and mt-tRNA$^{Lys}$. Cysteine is an essential amino acid in infants, due to the physiologically low activity of the cystathionine gamma-lyase (cystathionase) enzyme, and its low dietary availability in the first months of life contributes to reversible infantile hepatopathy (Sturman *et al*, 1970; Zeharia *et al*, 2009). Also, heteroallelic *EARS2* (mitochondrial glutamic acid tRNA synthetase, OMIM*612799) mutations show partial recovery of neurological and muscle symptoms around 1 year of age in two-third of patients (Steenweg *et al*, 2012; Talim *et al*, 2013).

In this study, we utilised unbiased genomic sequencing, transcriptomic and proteomic approaches to explore the reasons for the markedly reduced penetrance and define the molecular mechanism of the reversibility in RIRCD associated with the homoplasmic m.14674T>C mutation.

# Results

## Clinical presentation

We studied 42 individuals (27 affected, 15 unaffected) from 19 families from eight countries (UK, Germany, Sweden, Estonia, Italy,

USA, Canada, Brazil) carrying the homoplasmic m.14674T>C mt-tRNA$^{Glu}$ mutation (Table EV1, Fig EV1A). We present 10 new patients from eight families, while 17 patients from 11 families were reported previously (Houshmand *et al*, 1994; Horvath *et al*, 2009; Uusimaa *et al*, 2011; Joost *et al*, 2012). Informed consent was obtained from each participant approved by local ethics committees, and the study was approved by the Yorkshire & The Humber - Leeds Bradford (13/YH/0310). The skeletal muscle and fibroblast samples were stored in the Newcastle Biobank of the MRC Centre for Neuromuscular Diseases (Reza *et al*, 2017).

The clinical presentation of the patients and their family members homoplasmic for m.14674T>C is summarised in Table EV1. In brief, 22 patients (13 male/nine female) presented with muscle weakness before 3 months of age and showed partial or complete recovery between 4 and 30 months of age (RIRCD), or a mild, residual myopathy in 10 individuals. An additional four female homoplasmic mutation carriers from four different families did not present with weakness in the first months of life, however developed mild non-progressive muscle weakness at a later age. Fifteen homoplasmic mutation carriers were healthy and never showed myopathy or any other symptom which can be associated with mitochondrial disease. Two generations were affected in five families, with siblings in two families.

## Whole exome sequencing to search for common genetic modifiers of m.14674T>C

Whole exome sequencing (WES) was performed in 34 individuals (22 patients and 12 unaffected carriers). Mitochondrial DNA haplogroups were obtained from WES (Griffin *et al*, 2014), and the homoplasmic m.14674T>C mutation was detected in all individuals on twelve different haplogroup backgrounds, demonstrating that this mutation has arisen independently in unrelated families (Figs EV1B and EV2A). We did not identify any additional mtDNA variants in the affected individuals which could potentially affect mitochondrial translation and thus contribute to the clinical presentation.

Reversible infantile respiratory chain deficiency manifested with ~ 30% penetrance in the first year of life in previously reported families homoplasmic for m.14674T>C, raising the possibility of a genetic modifier (Boczonadi *et al*, 2015). We first hypothesised that a common genetic variant, either in affected patients or in healthy controls, may modify the clinical presentation of m.14674T>C(Horvath *et al*, 2009; Boczonadi *et al*, 2015) (Fig EV2B); however, no single variant segregated in all affected or unaffected individuals. We identified a frequent (minor allele frequency or MAF 0.4723, homozygote frequency 0.11) missense variant c.67C>T, p.Arg23Trp in *PDE12* which was homozygous in 14 affected individuals from 8 out of 19 families (with different mtDNA haplotype), although three unaffected homozygous carriers were subsequently found in one family (Affection vs *PDE12*, Fisher's Exact *P* = 0.087). PDE12 or 2′,5′-phosphodiesterase 12 is a major factor for the quality control of mitochondrial non-coding RNAs and the lack of PDE12 results in a spurious polyadenylation of the 3′ ends of the mitochondrial (mt-) rRNA and mt-tRNA (Pearce *et al*, 2017). The c.67C>T, p.Arg23Trp variant however, did not affect the respiratory chain complexes and the processing of 3′ poly(A) tails of the 16S mt-rRNA, because introducing the mutant gene rescued the phenotype of *PDE12* knockout cells (Fig EV3A–C).

In nine RIRCD patients from eight families (with different mtDNA haplotype), we detected the heterozygous common c.28G>T; p.Ala10Ser *TRMU* variant (Figs EV1A and EV2A), which has been shown to result in reduced thiouridylation of mt-tRNA[Glu] (Meng *et al*, 2017). It has a population total allele frequency of 0.09695 (gnomAD). Thus, the probability of carrying both p.Ala10Ser and m.14674T>C is approximately one in 100,000 ($1 \times 10^{-5}$), a frequency that is compatible with the rare occurrence of RIRCD in a subset of homoplasmic m.14674T>C carriers.

## Digenic inheritance of m.14674T>C with rare, damaging nuclear variants

Based on the frequency of m.14674T>C (0.01%) and the predicted 30% penetrance rate in previously reported families (Boczonadi *et al*, 2015), the estimated prevalence of RIRCD patients should be ~ 1/ 30,000 (0.003%), predicting > 2,000 RIRCD patients only in the UK. However, less than 100 RIRCD patients have been reported worldwide to date, suggesting that at least 100 times more unaffected individuals and families carry the homoplasmic m.14674T>C change. This is supported by the existence of two m.14674T>C carriers in a Danish type 2 diabetes study (Li *et al*, 2014) and two homoplasmic carriers in the NIHR BioResource Rare Diseases dataset. Heteroplasmic carriers (between 3 and 15% mutant allele) have also been found in three "1000Genomes" samples, and there are 11 carriers (nine females, two males) of the homoplasmic m.14674T>C in the UK Biobank European unrelated individuals ($N = 358,916$), confirming the MAF = $3 \times 10^{-5}$ (Wei *et al*, 2019). None of these carriers had any evidence of mitochondrial disease. The most likely scenario is that affected individuals carry another variant (or variants) in addition to m.14674T>C, and the additive effect of these variants may underlie the phenotype. Therefore, we filtered our WES data for damaging, rare variants specific to affected or unaffected individuals within the RIRCD pedigrees and detected 2,698 potentially damaging variants in affected and 1,067 variants in unaffected individuals within 19 families (Fig EV2B). No homozygous or compound heterozygous pathogenic variants were detected in genes involved in mitochondrial translation in any of the affected individuals. However, 6 different rare heterozygous damaging variants in *EARS2,* the mt-tRNA synthetase responsible for aminoacylation of mt-tRNA[Glu] co-segregated with the disease in 9 RIRCD patients from 6 families (Table EV1, Figs EV1 and EV2).

Three *EARS2* variants identified in our families (c.328G>A; p.Gly110Ser, c.670G>A; p.Gly224Ser and c.1547G>A; p.Arg516Gln) have been previously reported as pathogenic mutations in compound heterozygous state with other mutations in patients with autosomal recessive leukoencephalopathy with brainstem and thalamus involvement and high lactate (LBTL, OMIM *612799) (Steenweg *et al*, 2012) and affect the catalytic and anticodon binding domains of the protein (Moulinier *et al*, 2017) (Fig 1A). The other three heterozygous *EARS2* variants (c.358C>T; p.Arg120Trp, c.596A>G; p.Gln199Arg and c.263C>T; p.Ala88Glu) were predicted to be damaging by multiple Annovar annotation databases alter evolutionarily conserved amino acids, in the catalytic domain of the protein (Moulinier *et al*, 2017) (Fig 1A). The *EARS2* variants were all rare (MAFs < 0.002), and no homozygous individuals are reported in control databases (ExAC, gnomAD) except for c.670G>A; p.Gly224Ser (Table EV2). Despite the recent publications on patients with recessive *EARS2* mutations, clinical symptoms have not been observed to date in heterozygous carriers

of this gene (Steenweg *et al*, 2012). Based on the frequency of *EARS2* variants in international databases, we calculated that the cumulative frequency of carrying a heterozygous damaging variant in this gene is approximately 0.0135. Based on the 0.0001 MAF of m.14674T>C and 0.0135 cumulative MAF of heterozygous damaging variants in *EARS2*, the probability to carry both is approximately 1 in a million ($1.35 \times 10^{-6}$). This number is compatible with the extremely rare occurrence of RIRCD and supports the observation in our 6 families that both mt-tRNA[Glu] and *EARS2* variants contribute to the clinical manifestation of the disease.

A rare, damaging heterozygous *TRMU* variant (c.902A>G; p.Tyr301Cys) co-segregated with RIRCD in one family (Table EV1). It is also likely to contribute to the phenotype with a similar digenic mechanism. Additionally, a severely affected RIRCD patient carried heterozygous variants in both *EARS2* (c.328G>A; pGly110Ser) and *TRMU* (p.Ala10Ser) in addition to m.14674T>C, indicating a possible correlation between mutational load and disease severity. Mutations in *TRMU* either targeted the amino terminal catalytic domain (p.Ala10Ser) or the Carboxy terminal β-barrel domain (p.Tyr301Cys) (Numata *et al*, 2006), both occurring in amino acids that are not fully conserved across species (Fig 1A).

Seven affected individuals without heterozygous variants in *EARS2* or *TRMU* carried heterozygous variants in human disease genes interacting with mt-tRNA[Glu] such as *QRSL1* (another mt-tRNA synthetase aminoacylating mt-tRNA[Glu]), *GOT2* (mitochondrial glutamic-oxaloacetic transaminase) and *GLS* (mitochondrial glutaminase), both involved in glutamate or glutamine metabolism (Fig 1B) (Friederich *et al*, 2018; van Karnebeek *et al*, 2019; van Kuilenburg *et al*, 2019). All these mutations affect the catalytic domains of these proteins (Nakamura *et al*, 2006; Han *et al*, 2011) (https://www.uniprot.org/) and highly conserved amino acids (Fig 1A). One patient carried a rare heterozygous frameshift variant in the gene *MSS51*, which has been shown to act as a translation activator of *MTCO1* and an assembly factor of cytochrome *c* oxidase (Moyer & Wagner, 2015; Garcia-Villegas *et al*, 2017), and had a significantly decreased gene expression in the affected patients' muscle (family 18; Fig 1B, Table EV2). Together the cumulative MAFs of damaging *EARS2, TRMU, QRSL1, GOT2, GLS* and *MSS51* variants found in the ExAC database (MAF = 0.32) could explain the affected status of 32% of m.14674T>C mutation carriers in our cohort, which is consistent with previous observation that ~ 30% of homoplasmic mutation carriers within RIRCD families develop the symptoms (Boczonadi *et al*, 2015). The cumulative incidence of damaging variants in *EARS2, TRMU, QRSL1, GOT2, GLS, MSS51* and m.14674T>C (Fig EV1C) shows a significant increase in the mean number of variants in these genes between RIRCD affected and unaffected carriers (mean alleles: affected = 2.29, unaffected = 1.33; $t = 4.77$; $P = 5.1 \times 10^{-5}$) and even more between RIRCD and controls (mean alleles: affected = 2.29, 1000 Genomes = 0.73; $t = 10.89$; $P = 3.1 \times 10^{-10}$). None of the unaffected UK Biobank m.14674T>C carriers are heterozygous for pathogenic variants in *EARS2* or *TRMU*.

## Transcriptomic and proteomic changes in skeletal muscle biopsies identify a metabolic shift and increased mitochondrial biogenesis modulated by ISR

We studied gene expression in the skeletal muscle of six RIRCD patients biopsied during the symptomatic phase and six healthy

**A**

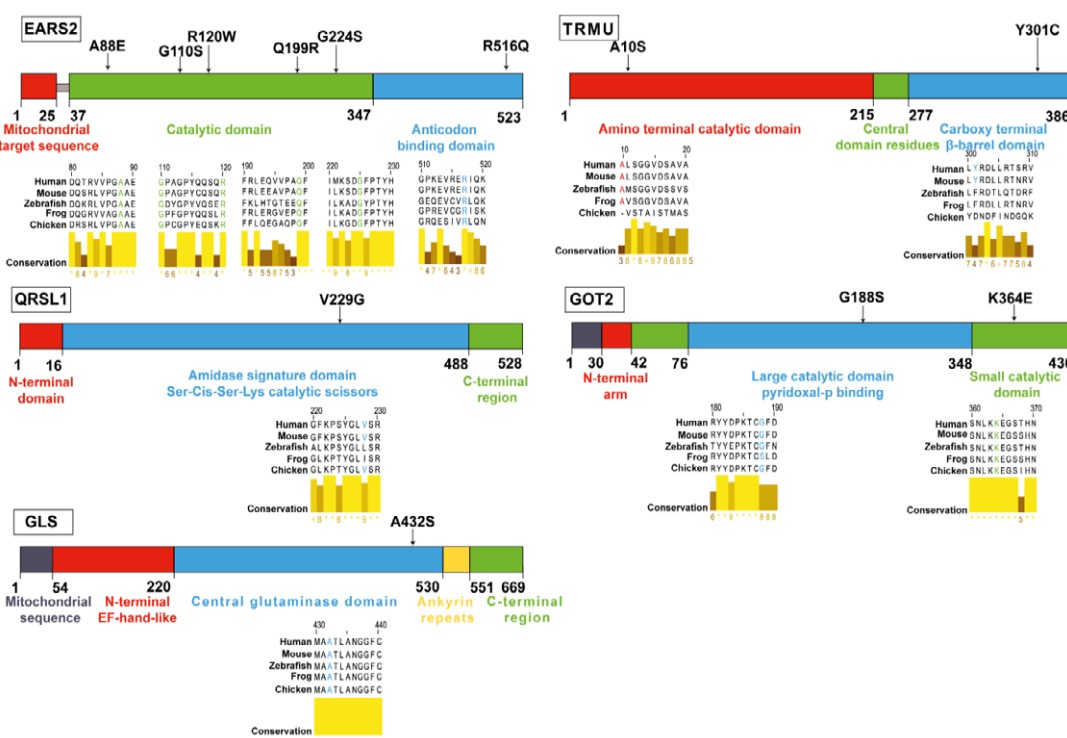

**B**

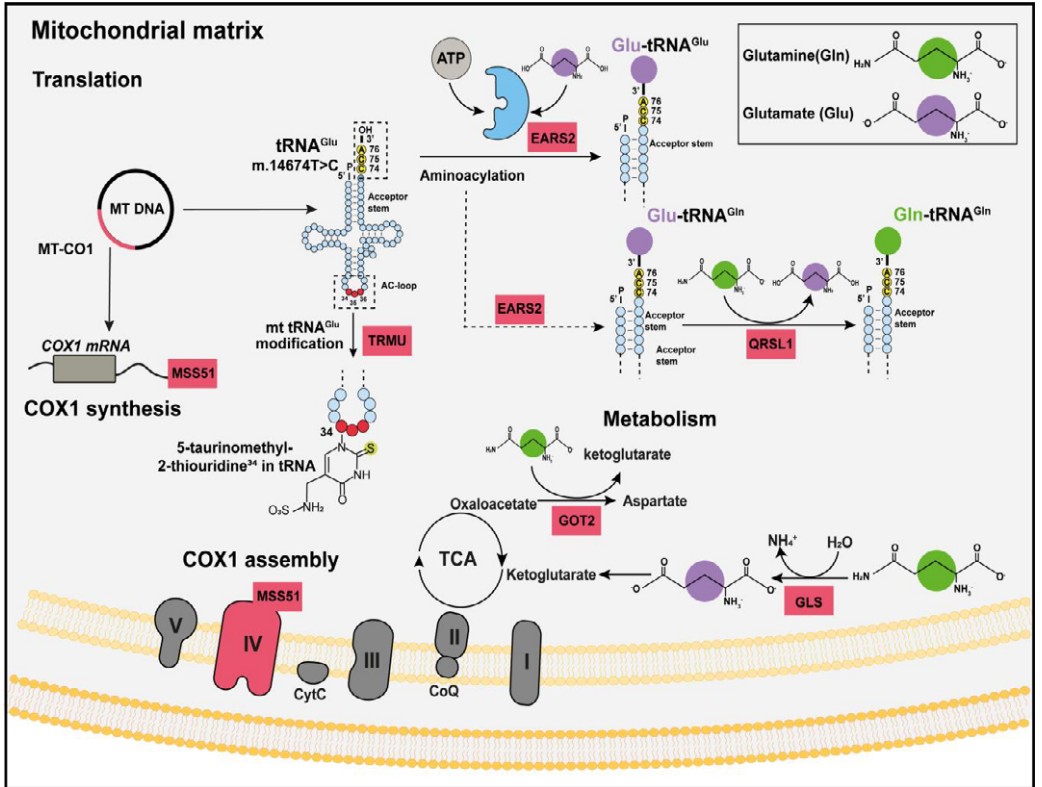

**Figure 1.**

**Figure 1.  Distribution of the identified mutations in nuclear encoded proteins in our RIRCD cohort and schematic illustration of the interplay between affected proteins.**

A  Schematic diagram of affected proteins and the locations of the point mutations identified in: EARS2 p.Gly110Ser, p.Gly224Ser, p.Arg516Gln, p.Arg120Trp, p.Gln199Arg and p.Ala88Glu mutations-catalytic and anticodon binding domain of the protein; TRMU p.Tyr301Cys and p.Ala10Ser- amino terminal catalytic and carboxy terminal β-barrel domain; QRSL1 p.Val229Gly-amidase signature domain; GOT2 p.Gly188Ser and p.Lys364Glu located in the large catalytic and small catalytic domains and GLS p.Ala432Ser found in the central glutaminase domain. Multiple sequence alignment of the above mentioned proteins shows the conservation of the affected amino acids across species using Jalview software. Analysis was performed using protein sequences from human (Q5JPH6, O75648, Q9H0R6, P00505, O94925), mouse (Q9CXJ1, Q9DAR5, Q9CZN8, P05202, D3Z7P3), zebrafish (Q0P499, Q503J2, F1QAJ4, Q7SYK7, Q8JFS4), frog (Q66JG3, F6TJB0, Q0VFI5, Q28F67, F7B417) and chicken (Q5ZJ66, Q5ZKW0, F1NLA0, P00508, A0A1D5PNV1).

B  Schematic diagram of affected proteins showing their contribution to (i) mitochondrial protein translation (EARS2 which aminoacylates mt-tRNA$^{Glu}$ and mt-tRNA$^{Gln}$ and QRLS1 which transamidates Glu-tRNA$^{Gln}$ to form the correctly charged Gln-tRNA$^{Gln}$), (ii) amino acid metabolism (GOT2 which catalyses the interconversion of oxaloacetate and glutamate into aspartate and α-ketoglutarate while GLS metabolises glutamine to ammonia and glutamate which is further catabolised to the TCA intermediate α-ketoglutarate to fuel the mitochondrial carbon pool) and (iii) COX1 assembly and synthesis due to the dual role of MSS51. MSS51 is not included in (A) as the identified mutation in our patients is a frameshift mutation leading to a premature stop codon and most likely no protein being translated.

age-matched controls. Two of the RIRCD patients carried digenic mutations in *EARS2* (Table EV1). Differential expression analysis in RIRCD biopsies identified 1,398 RNAs with significant (*P*-adjusted ≤ 0.05) overexpression, and 1,905 RNAs with significant under-expression compared with control biopsies (Fig 2A–C). We also analysed consecutive biopsies from one patient during the affected phase and after recovery, indicating metabolic alterations in RIRCD, which reversed in parallel with the clinical recovery (Fig 2D). Comparative proteomic analysis of muscle protein extracts derived from three RIRCD patients (Table EV1) vs three controls (3 months, 18 years, 19 years) led to the identification of 1,600 proteins from which 141 were statistically significantly upregulated and 134 downregulated (*P*-Anova ≤ 0.05) in RIRCD muscle (Figs 2E and F, and EV4). Changes in mtDNA-encoded subunits and key proteins identified in our analysis were confirmed by immunoblotting (Fig 3A and B). Integration of transcriptomic (*genes - italic*) and proteomic (proteins - normal) data was performed to highlight metabolic pathways, involved in the molecular mechanism of RIRCD.

Fibroblast Growth Factor 21 (*FGF21*), a hormone with crucial role in regulating lipid and glucose metabolism (Forsstrom *et al*, 2019) and *GDF15*, both biomarkers of mitochondrial translation deficiencies were profoundly increased in RIRCD muscle and normalised after recovery (Figs 2D and 3F). Whether *FGF21* and *GDF15* are drivers of the observed metabolic shift in RIRCD patients, or just by-standing biomarkers is currently unclear. Our data are also consistent with the activation of a mitochondrial integrated stress response (ISR$^{mt}$) as besides these two cytokines we observed an increase in other transcripts belonging to these pathway such as activating transcription proteins or ATFs such as *ATF5* and *ATF3*, as well as tribbles homolog 3 *TRIB3* and DNA damage-inducible transcript protein 3 *DDIT3* (Khan *et al*, 2017; Forsstrom *et al*, 2019) (Figs 2A and 3F). While *ATF3* and *DDIT3* are also components of the UPR$^{mt}$, we did not observe any additional changes in heat shock proteins such as *HPS60*, *HPS10* or *HPS20*, similar finding being reported in Deletor mice (Forsstrom *et al*, 2019) We also detected alterations in genes associated within serine biosynthesis (*PSAT1, PHGDH* and *SDSL*), one carbon metabolism (*MTHFD1L, SHMT1, SHMT2*) and methionine cycle (*AHCY, BHMT2* and *CHDH*; Fig 3F), also components of the ISR$^{mt}$ (Forsstrom *et al*, 2019). Intriguing most of the transcripts corresponding to ISR return to basal levels or decrease in recovered muscle (Figs 2D and 3F).

Our analysis showed also alterations in other metabolic enzymes. An increase in pyruvate dehydrogenase kinase isozyme 2 (PDK2), SUCLG2 and *ALDH6A* suggests a redirection of the metabolic flux

from the respiratory chain to TCA, suggesting an increase in this metabolic pathway (Majer *et al*, 1998) and further supported by limited conversion of TCA components into glucose, illustrated by decreased *PCK2* (Park *et al*, 2018) (Fig 2, Datasets EV1 and EV2). Enzymes involved in the oxidation of fatty acids were upregulated in RIRCD muscle (ACADM, ACADS, ACAA2, ACOT9/*ACOT9*, HCD2, ACDSB), including enzymes that use short- (ACSS1/*ACSS1*), medium- and long-chain fatty acids (ACADVL, ACSF2 and DHB12) in order to contribute to the acetyl-CoA pool, which is further channelled into the TCA (Forster & Staib, 1992; Kiema *et al*, 2014; Tillander *et al*, 2014) (Fig 2 and Datasets EV1 and EV2). In addition, ACAT1/*ACAT1* was decreased in patient muscle, slowing the conversion of acetyl-CoA into acetoacetyl CoA and CoA, thus maintaining the cellular pool of acetyl-CoA needed for the TCA (Haapalainen *et al*, 2007). All of this changes show that we have a strong metabolic shift from OXPHOS towards fatty acid oxidation and TCA cycle. We hypothesise that this metabolic remodelling acts as a compensatory mechanism in order to overcome the energetic deficit caused by blocked oxidative phosphorylation.

In addition, in our omics analysis we detected changes consistent with an activation of mammalian target of rapamycin (mTOR) and PI3K/AKT (phosphatidylinositol 3-kinase) in patient muscle (Figs 2C and 3F and Datasets EV1 and EV2). Although we do not see any changes in mTOR transcript, we did observe in our data a decrease in *DEPTOR*, a known natural inhibitor of mTOR (Peterson *et al*, 2009) in affected muscles compared with controls while during recovery this transcript shows a significant increase (Fig 3F). Additionally, we observed an increase in several mTOR targets in our data, such as *RRAGC* (cellular response to amino acid availability), *EIF4EBP1* (protein translation) and *PPARG* (lipid synthesis; Laplante and Sabatini 2009) in RIRCD compared with controls (Fig 3), while in recovered patients these transcripts decrease (Fig 3F) and enhanced expression of *PIK3R3, PIK3CD* and *PIK3CG* in both affected and recovered muscle (Fig 2A). Moreover, we saw an increase in *PGC1α* (Fig 3F), a common target of mTOR which controls mitochondrial biogenesis (Cunningham *et al*, 2007), illustrated here, by a significant increase of mtDNA copy numbers at recovery (Fig 3C).

**Amino acid regulation of the mitochondrial respiratory chain may provide a muscle-specific rescue mechanism in RIRCD**

Our data revealed that several pathways involved in the catabolism of amino acids are increased in patient muscle (Fig 2). We observed

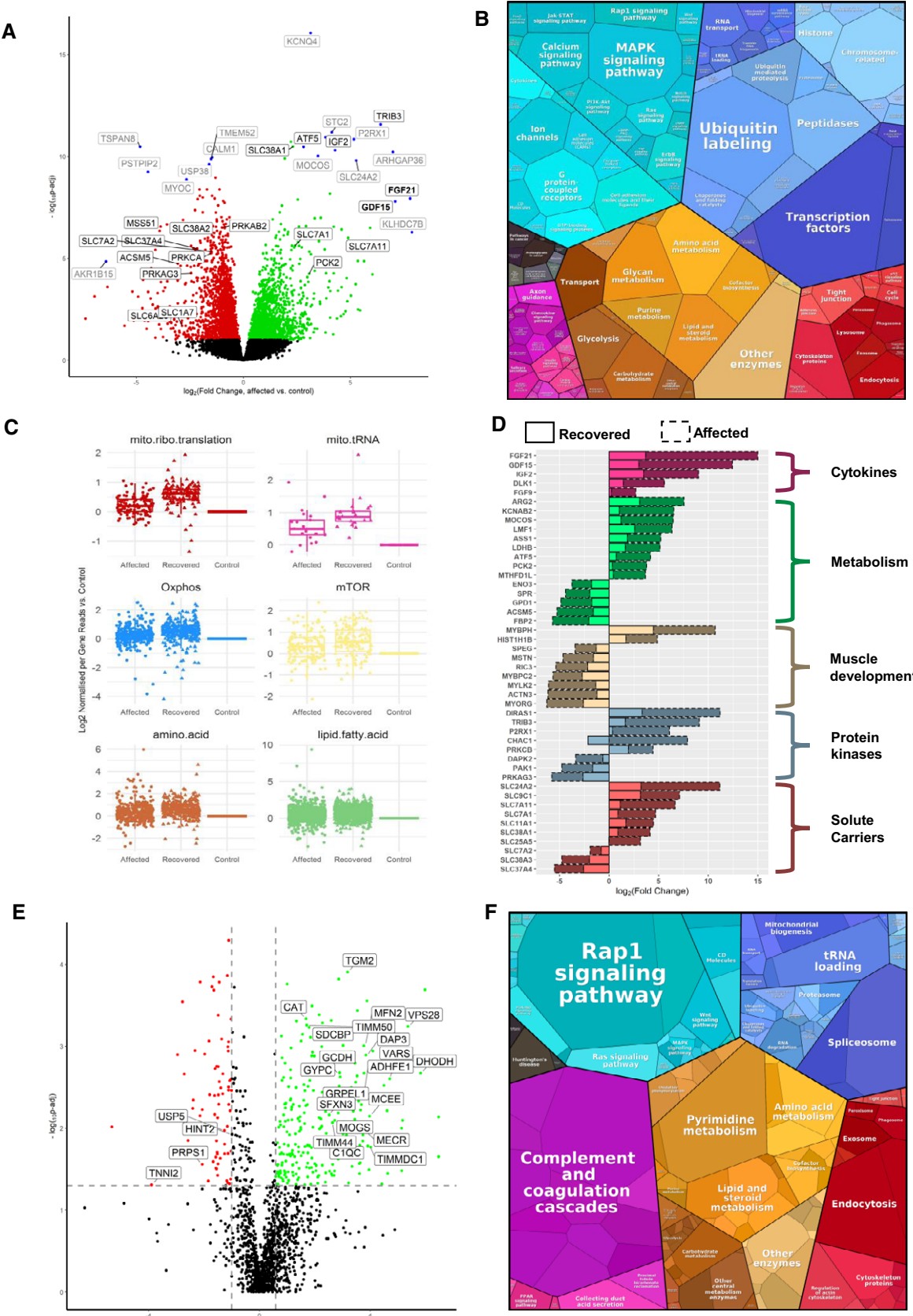

Figure 2.

**Figure 2.  RNA-seq differential expression (A-C) and Proteomics analysis (E&F) in human muscle.**

A   Volcano plot shows genes where RNA reads are statistically significant under- or over-expressed in five RIRCD affected individuals vs six controls resulted from DESeq2 analysis. Decreased genes are represented in red, the upregulated ones are in green while blue dots highlight most outlying genes (defined by absolute ($\log_2$(fold change) + $\log_{10}$(Padj)) > 10). Genes that were not statistically significant are represented in black. Grey and black gene names indicate top 30% of the most dysregulated genes with a highlight on ones involved in metabolism. For the calculation of *P*-value or *Padj*, we have employed the R package DESeq2 where the *P*-values obtained by the Wald test are corrected for multiple testing using the Benjamini and Hochberg method by default.

B   Representation of the KEGG terms associated with differentially expressed RNAs and their corresponding proteins in five affected patients vs six controls showing major pathway changes in patient muscle (https://bionic-vis.biologie.uni-greifswald.de/).

C   Abnormal transcription of genes involved in various biological metabolic pathways derived from our RNA-seq (RNA-seq read counts/person/gene for *n* = 6 RIRCD affected individuals, *n* = 5 controls and *n* = 1 (biological replicate analysed in four technical replicates) recovered) data showing that their expression returned close to normal levels in patient muscle after recovery. The central band represents the median, the lower and upper hinges correspond to the first and third quartiles (25 and 75%) while the whiskers extend to the highest and lowest points within the data (1.5× the inter-quartile range).

D   Relevant functional changes in gene expression showing the major genes that return to baseline or contribute to the recovery in RIRCD patients. The data depicted are between the affected and recovered biopsies of the same patient (F6/1M).

E   Comparative proteome profiling of proteins that are significantly under- or over-expressed in RIRCD affected vs control individuals (*P*-Anova < 0.05-statistically significant - horizontal line). Proteins which are decreased are represented in red while the upregulated ones are in green while proteins that did not reach statistical significance threshold are marked in black. Vertical lines are delimiting the regulated proteins resulted from the proteomics analysis (the cut-off values were determined based on the 2× standard deviation and the normal distribution from all identified protein's $\log_2$ ratio in which the bell curve is symmetric around the mean, therefore, an average $\log_2$ ratio of a protein which < −2.2 or > 0.98 was considered as regulated).

F   Proteomap representation of the major altered pathways in RIRCD affected muscle compared to control individuals where the size of each circle or hexagon represents the fold change.

a significant increase in GCN2, a serine/threonine kinase activated by amino acid deprivation (Fig 3D and E), in affected muscle concomitant with a significant increase in ATF4 a downstream target of GCN2 in RIRCD muscle (Fig 3D and E), which represses general protein translation concomitant with stimulating the expression of amino acid biosynthetic and transporter genes (Masson, 2019). Moreover, these two proteins have been described to be major components of ISR^mt in cultured cells (Quiros *et al*, 2017; Forsstrom *et al*, 2019).

An increase in proteins that act as transporters of different amino acids (*SLC38A1, SLC1A7, SLC7A11/SLC7A11, SLC7A1,* TIM13, TIM50, TIM44, GRPE1) (Kandasamy *et al*, 2018) was observed in affected muscle (Figs 2 and 3D–F and Datasets EV1 and EV2). These proteins are sensitive to amino acid status and are regulated in response to cellular needs. The increased need for amino acids in RIRCD muscle is further supported by notable increase in genes involved in the metabolism of alanine, aspartate and glutamate (*BCAT1, FOLH1, CAD, ASNS, ASS1*). The metabolism of branched chain amino acids such as valine, isoleucine, lysine (*HMGCS2, GCDH, DGLUCY, ADHFE1*) (Zhang *et al*, 2017) and the degradation of lysine (GCDH) (Sauer, 2007) and tryptophan (*TPH1, KYNU*) (Pabarcus & Casida, 2002) was also significantly increased in RIRCD (Fig 2 and Datasets EV1 and EV2).

**Cysteine and glutamic acid depletion triggered combined respiratory chain deficiency in fibroblasts with m.14674T>C**

Cysteine has been suggested to be an essential amino acid in the first months of life, and its limited availability in infants may explain the reversible liver presentation of *TRMU* mutations (Zeharia *et al*, 2009) and potentially further compromises mt-tRNA^Glu steady state in RIRCD (Boczonadi *et al*, 2013). Glutamic acid is the cognate amino acid added by EARS2 to mt-tRNA^Glu. Therefore, to understand if cysteine or glutamic acid availability influences mitochondrial translation in RIRCD, we used primary fibroblasts of affected and unaffected mutation carriers of m.14674T>C and controls. These fibroblasts did not show respiratory chain deficiency in regular medium (Boczonadi *et al*, 2013). Lower cysteine levels in the cell

culture medium (0.02 mM cysteine) for 12 days had a deleterious effect on complex I and complex IV in all fibroblasts, with an enhanced effect in cells from patients carrying the m.14674T>C mutation (Fig 4A and B). In contrast, glutamine and glutamic acid depletion for 12 days triggered a mild, but significant decrease in MTCO1 and MTCO2 (Complex IV; Fig 4C and D) in fibroblasts of patients with digenic heterozygous *EARS2* mutations. The observed changes were less severe than after cysteine depletion as glutamine and glutamic acid are considered to be non-essential and easily synthetised via the TCA (Liaw & Eisenberg, 1994).

Aminoacylation of mt-tRNA^Glu was not significantly altered in fibroblasts with digenic mutations compared with healthy homoplasmic carriers of m.14674T>C (Fig EV5), while, studying aminoacylation in skeletal muscle was not possible due to the limited size of the biopsies.

**Digenic mutations in m.14674T>C and EARS2 result in a more severe defect of mtDNA-encoded gene expression in muscle**

We compared differential expression of two RIRCD patients with digenic *EARS2* mutations (F2/1M, F5/1M) to two RIRCD patients without digenic *EARS2* mutations (F2/1M, F5/1M) and identified significantly altered (*P*-adjusted < 0.05) gene expression of 59 mRNAs (36 increased, 23 decreased) including complex I proteins such as *ND1, ND2, ND4L, ND5, ND6* (Fig 5A–C). Skeletal muscle of RIRCD patients with digenic *EARS2* mutations showed lower expression of mtDNA-encoded genes. Furthermore, the degree of decrease in mRNA counts correlates directly with the proportion of glutamine and glutamic acid residues of the mtDNA-encoded proteins (*r* = 0.61677, *P* = 0.024), with most severely affected subunit being NADH dehydrogenase, which contains the highest number of these amino acids (Fig 5B).

## Discussion

We studied 42 individuals from 19 families homoplasmic for m.14674T>C, and in 24 of 27 affected individuals, we identified

additional heterozygous damaging variants in *EARS2* and in four other nuclear genes that modify or aminoacylate mt-tRNA$^{Glu}$ and mt-tRNA$^{Gln}$ (*TRMU, QRSL1*) or are involved in glutamine/glutamate metabolism (*GOT2, GLS*; Fig 1). These variants were absent in

healthy carriers of m.14674T>C. Three of the nine *EARS2* variants detected in RIRCD patients were previously reported in patients with autosomal recessive leukoencephalopathy with thalamus and brainstem involvement and high lactate (LTBL) (Steenweg *et al*, 2012).

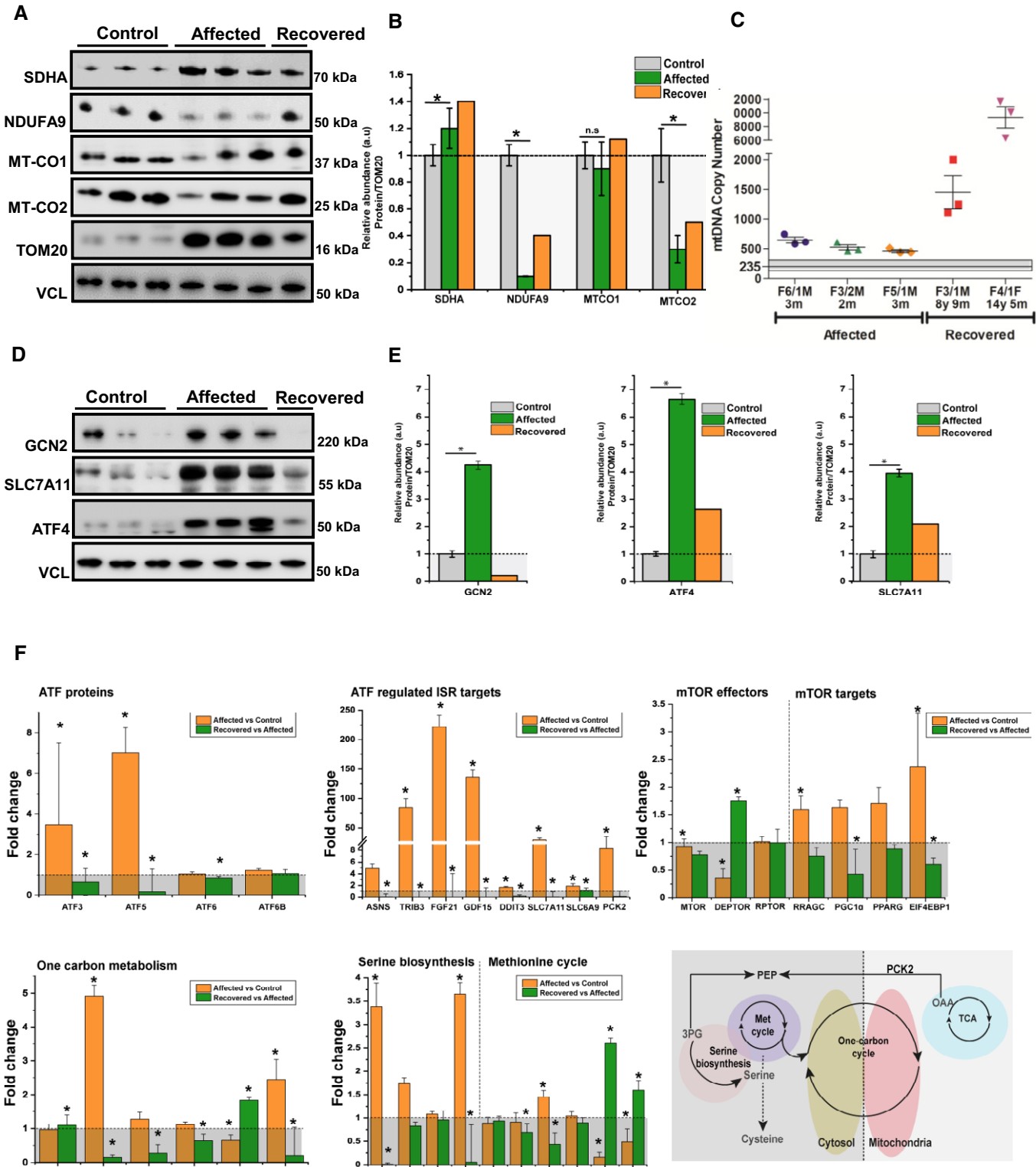

Figure 3.

**Figure 3.  Immunoblotting in skeletal muscle of patients and transcriptomics detected alterations of several mitochondrial complex subunits, the activation of integrated stress response and mTOR.**

A  Western blotting analysis depicts mitochondrial proteins (NDUFA9-Complex I, SDHA-Complex II, MTCO1-Complex IV and MTCO2-Complex IV) levels in control-derived muscle (*n* = 3), patient-derived muscle during the symptomatic phase (*n* = 3) and patient-derived muscle after recovery (*n* = 1). VCL was used as a loading control while the densitometry analysis was done based on the mitochondrial protein TOM20.

B  Densitometry analysis of the immunoblotting from (A) showing an increase in SDHA (Complex II), while NDUFA9 (Complex I) probably as a compensatory mechanism and MT-CO2 (Complex IV) show significant reductions as the majority of mitochondrial encoded proteins belong to this two complexes, thus being destabilised by the defective mitochondrial mistranslation. Graphs show means ± SD of triplicate samples (controls and affected muscle), and results obtained from (*n* = 1) recovered muscle. For statistical analysis, unpaired Student's *t*-test was employed where *\*P* ≤ 0.05 were considered statistically significant.

C  Mitochondrial DNA copy numbers were determined in affected (*n* = 3) and recovered individuals (*n* = 2) and compared with control individuals (*n* = 4, 3 months, 3 years, 4 years and 7 years) using qPCR showing a significant increase in recovered muscle. Graph shows mean ± SD for each biological replicate. The average of controls are represented as horizontal line, and graphs show mean ± SD of replicates.

D  Western blotting analysis of proteins involved in the amino acid sensing and integrated stress response: GCN2, SLC7A11 and ATF4 in control-derived muscle (*n* = 3), patient-derived muscle during the symptomatic phase (*n* = 3) and patient-derived muscle after recovery (*n* = 1). All samples were normalised to VCL which was used as a loading control.

E  Densitometry analysis of proteins from (D) showing a significant increase of major transducers of ISR such as GCN2 and ATF4. Consistent with ISR, activation is also the increase in SLC7A1. Graphs show means ± SD of triplicate samples (controls and affected muscle), and results obtained from *n* = 1 recovered muscle. Similar results were observed in at least two independent results (due to the limited amount of sample). Statistical analysis unpaired Student's *t*-test was employed where *\*P* ≤ 0.05 was considered statistically significant.

F  Transcriptomic pathways related to ATFs and ISR changed in affected patients compared with controls (RIRCD *n* = 5; controls *n* = 6) and in recovered muscle compared with affected muscle (*n* = 1, affected and recovered muscle was collected from the same individual and was analysed using 4 technical replicates) and schematic representation of the metabolic pathways impacted in patient muscle (bottom, right panel). Results are expressed as fold change of the affected compared with controls or of the affected muscle compared with recovered and data are shown as mean ± SEM. For the calculation of *P*-value or *P*adj, we have employed the R package DESeq2 where the *P*-values obtained by the Wald test are corrected for multiple testing using the Benjamini and Hochberg method by default where *\*P* ≤ 0.1.

Source data are available online for this figure.

While heterozygous pathogenic *EARS2* mutations do not lead to clinical symptoms, the co-occurrence of these variants with m.14674T>C in infants was associated with RIRCD (Table EV1), and their disruptive effect on gene expression was demonstrated in skeletal muscle biopsies (Fig 5). The catalytic function of *EARS2* is to aminoacylate mt-tRNA$^{Glu}$ at the 3′ end (m.14674T>C) (Steenweg *et al*, 2012); therefore, the co-occurrence of these variants may further compromise the aminoacylation and the steady state of mt-tRNA$^{Glu}$ leading to a digenic inheritance in RIRCD. However, fibroblasts carrying digenic mutations (m.14674T>C and *EARS2*) did not show significantly reduced aminoacylation of mt-tRNA$^{Glu}$, compared with fibroblasts of healthy carriers of m.14674T>C (Fig EV5). It is possible that aminoacylation of mt-tRNA$^{Glu}$ would be more affected in skeletal muscle; however, we could not study this in frozen biopsies. Additionally, EARS2 can have also non-canonical functions (e.g. amino acid sensing and regulation) thus the heterozygous mutations seen in RIRCD patients could alter these and worsen the effect of m.14674T>C. The other co-segregating nuclear variants may also affect aminoacylation of mt-tRNA$^{Glu}$ and mt-tRNA$^{Gln}$ (*QRSL1*) or glutamic acid and glutamine levels (*GOT2, GLS*) in mitochondria (Fig 1B and Table EV2). Nine RIRCD patients carried a common *TRMU* variant (p.Ala10Ser), previously shown to interact with m.1555A>G to cause deafness (Meng *et al*, 2017). *TRMU* downregulation in cells together with m.14674T>C alters mt-tRNA$^{Glu}$ steady state (Boczonadi *et al*, 2013), suggesting again a digenic mechanism if these variants co-occur. In one family, the index patient carried a variant in *MSS51* (Bareth *et al*, 2016; Garcia-Villegas *et al*, 2017), a protein that regulates translation and assembly of cytochrome *c* oxidase (Fig 1 and Table EV1). The prevalence of the co-segregating nuclear mutations may explain the model of only one out of ~ 100 m.14674T>C mutation carriers expressing symptoms (Table EV2). True digenic inheritance has been reported in some other genetic conditions (facio-scapulo-humeral muscular dystrophy—FSHD, kidney diseases, deafness,

etc.) (Nagara *et al*, 2018); however, RIRCD is the first example of a digenic mtDNA–nuclear DNA interaction.

We previously reported that > 30% of normal mt-tRNA$^{Glu}$ steady state in healthy carriers of homoplasmic m.14674T>C is sufficient to maintain mitochondrial protein synthesis in skeletal muscle, while individuals < 15–20% mt-tRNA$^{Glu}$ were affected (Horvath *et al*, 2009). Here, we show that the co-occurrence of heterozygous *EARS2* (or other nuclear variants with similar function) further compromise the fragile compensatory rescue mechanisms in infants and act as a second hit, explaining the clinical manifestation of RIRCD. Transcriptomic and proteomic signature of RIRCD skeletal muscle biopsies detected profoundly increased *FGF21* and *GDF15*, components of the ISR and significant changes in several components signalling a metabolic shift from OXPHOS towards lipid metabolism and the TCA cycle, while glycolysis was reduced (Figs 2 and 3F). In support of our findings, increased CK, low carnitine, slightly increased acyl-carnitines, increased TCA cycle metabolites and accumulation of lipids, glycogen and abnormal mitochondria have been reported previously in RIRCD patients (Boczonadi *et al*, 2015).

*FGF21* has been recently described to drive the integrated mitochondrial stress response and activate a cascade of events in mitochondrial myopathy patients and mice, thus leading to a distinct metabolic remodelling (von Holstein-Rathlou *et al*, 2016; Lehtonen *et al*, 2016; AlJohani *et al*, 2019; Forsstrom *et al*, 2019). Here, we show that *FGF21* is associated with the activation of an integrated stress response (increased ATFs, especially ATF5, GCN2, amino acid transporters like *SLC7A11*/SLC7A11, one carbon metabolism and serine biosynthesis) (Pakos-Zebrucka *et al*, 2016; Forsstrom *et al*, 2019) in RIRCD, concomitant with an increase in fatty acid oxidation and TCA cycle associated enzymes, thus, leading to an alternative energy source (bypassing complex IV and V), most likely to compensate for the lack of ATP. Our data based on the investigation of a limited number of muscle biopsies suggest that the alleviation of the mitochondrial ISR along the disease is associated with the

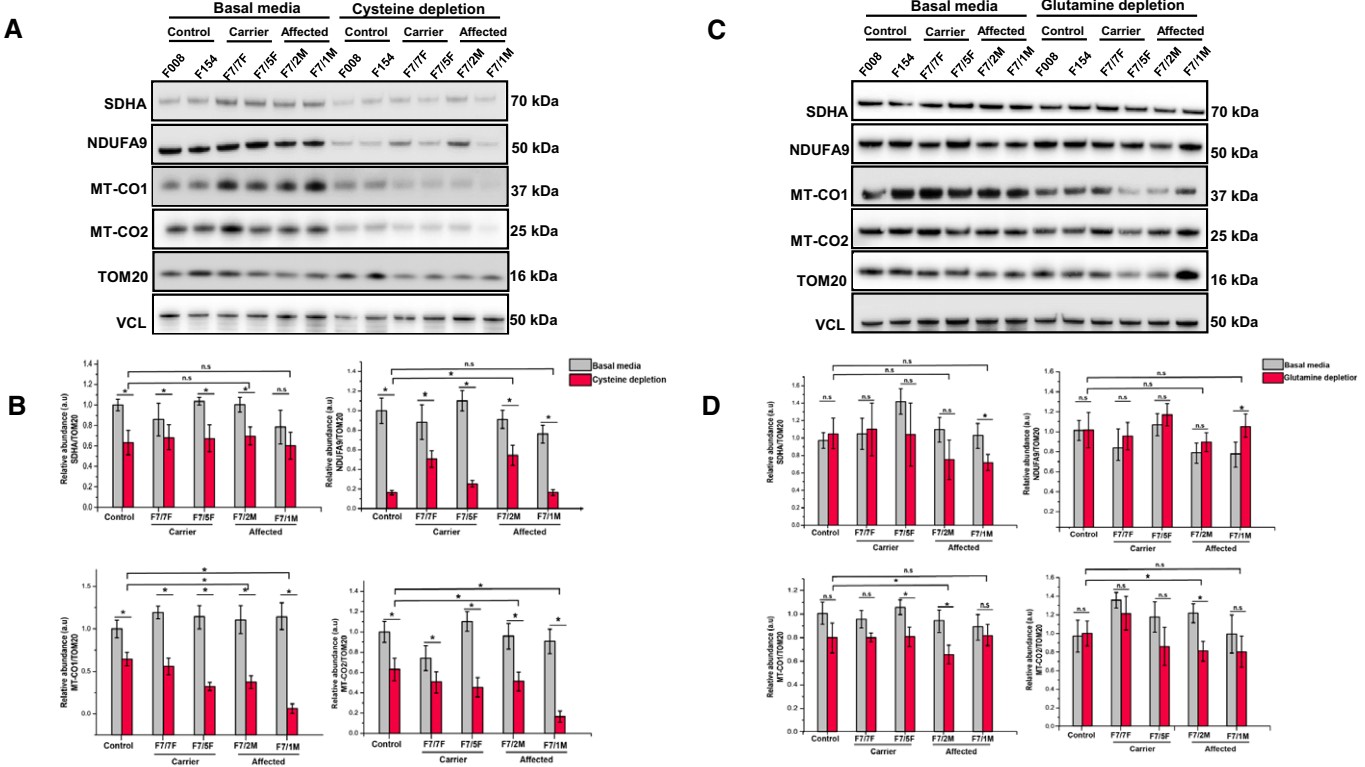

**Figure 4. Cysteine depletion alters expression of complex I and IV in RIRCD fibroblasts.**

A  Patient n = 4 (F7/7F-healthy carrier, F7/2M-RIRCD, F7/5F-healthy carrier, F7/1M-RIRCD without a clear second mutation) and control fibroblasts (n = 2) were grown for 12 days in media containing 0.02 mM cysteine and 5% dialysed FBS. Proteins belonging to mitochondrial complexes (NDUFA9-Complex I, SDHA-Complex II, MTCO1-Complex IV and MTCO2-Complex IV) were analysed via Western blotting. VCL was used as a loading control, and the densitometry analysis was done based on the mitochondrial proteinTOM20.

B  Densitometry analysis of proteins from (A) showing no significant changes in OXPHOS proteins upon glutamine/glutamic acid depletion. Control bar represents an average of the two control fibroblast lines employed (F008 and F154). Graphs show means ± SD of duplicate samples. For statistical analysis, unpaired Student's t-test was employed where *P ≤ 0.05 was considered statistically significant.

C  Patient (F7/7F-healthy carrier, F7/2M-RIRCD, F7/5F-healthy carrier, F7/1M-RIRCD without a clear second mutation) and control fibroblasts were grown for 12 days in media with no added glutamine or glutamic acid and 5% dialysed FBS. Cells were lysed, and the above mentioned mitochondrial proteins were checked via Immunoblotting. VCL was used as a loading control while the densitometry analysis (D) was done based on the mitochondrial proteinTOM20. Control bar represents an average of the two control fibroblast lines employed (F008 and F154). Although we could observe significant changes upon cysteine depletion in all conditions, the phenotype was exacerbated in the digenic fibroblasts compared with the controls. Graphs show means ± SD of duplicate samples (*P ≤ 0.05 unpaired Student's t-test was employed). All experiments were performed in triplicate n = 3, and similar results were observed in all experiments.

Source data are available online for this figure.

clinical recovery (Fig 3F). In contrast to progressive mitochondrial myopathies that have a long term pathogenic ISR, RIRCD patients present with a milder short term ISR as we observed a significant decrease in related proteins in recovered muscle (Fig 3D–F) which induces a beneficial hormetic response by activating protective cellular mechanisms that aid cells against amino acid deprivation, metabolic insults and oxidative stress (Pakos-Zebrucka et al, 2016). Additionally, this suggests that FGF21 and GDF15, which are biomarkers of mitochondrial diseases, could be used to follow treatment responses and disease progression/recovery in patients with mitochondrial disease.

We suggest that the events in RIRCD muscle take place in three distinct phases (Fig 6). In phase 1, the translational defect together with a limitation of amino acids results in activation of the mitochondrial integrated stress response (ISRmt), with increased FGF21 and GDF15, activation of GCN2 and induction of ATF5 and ATF4

leading to the inhibition of general cytosolic protein synthesis and increases the translation of specific proteins such as adaptive genes that are involved in maintaining the cellular homeostasis (amino acid transporters, metabolic enzymes) (Melber & Haynes, 2018; Xia et al, 2018). This in turn helps to conserve nutrients thus providing a temporary relief from metabolic stress (Ron, 2002).

Our transcriptomic data are in agreement with previous evidence that points to a clear contribution of *ATF5* to RIRCD ISRmt in human skeletal muscle (Forsstrom et al, 2019). The role of ATF4 in muscle during ISRmt has not been verified in human mitochondrial muscle disease, only in cultured cells (Quiros et al, 2017) and in mouse muscle where ATF4 activation is FGF21 dependent (Forsstrom et al, 2019). While ATF4 has been suggested to be regulated by an ISRmt independent mechanism in proliferating cells (Forsstrom et al, 2019), we hypothesise that in skeletal muscle of RIRCD patients ATF4 is sensitive to the amino

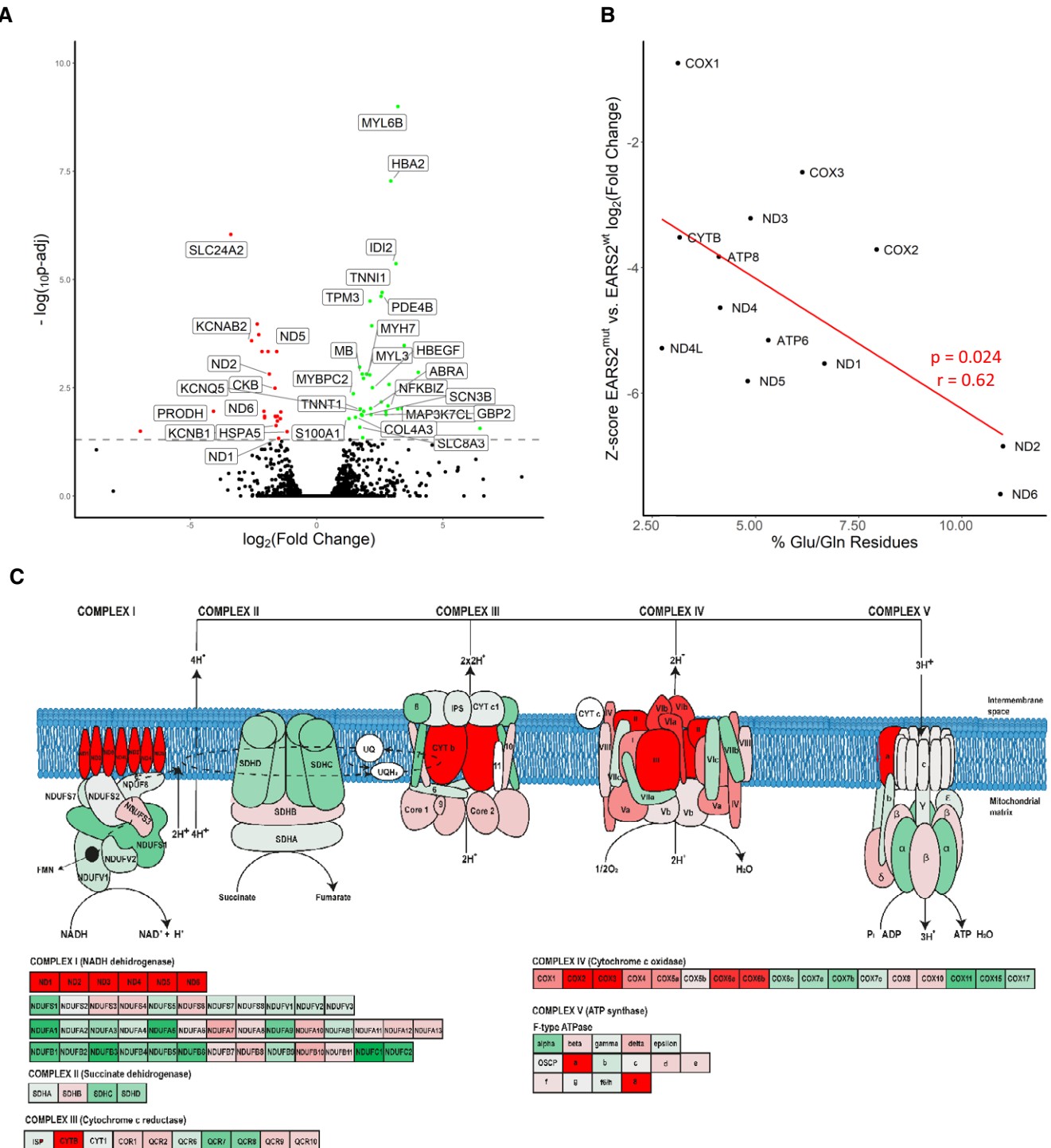

**Figure 5. Differential gene expression of RIRCD muscle biopsies with digenic mutations (m.14674T>C plus heterozygous *EARS2*) compared with RIRCD muscle (m.14674T>C), without *EARS2* mutations.**

A Volcano plot analysis of transcriptomic changes where statistically significant genes are delimited by a horizontal line. Decreased genes are represented in red while the upregulated ones are in green, while genes that were not statistically significant are represented in black. For the calculation of *P*-value or *P*adj, we have employed the R package DESeq2 where the *P*-values obtained by the Wald test are corrected for multiple testing using the Benjamini and Hochberg method by default. Labelled gene names represent top 60% most regulated genes.

B Analysis of altered mitochondrial gene expression resulted from our RNA-seq showing that affected genes and their coding proteins carry a high numbers of glutamic acid and glutamine residues, in accordance with the mitochondrial translational defect.

C Schematic representation of the mitochondrial respiratory chain enzymes resulted from the differential gene expression where red colour means significantly downregulated, green means significantly upregulated genes.

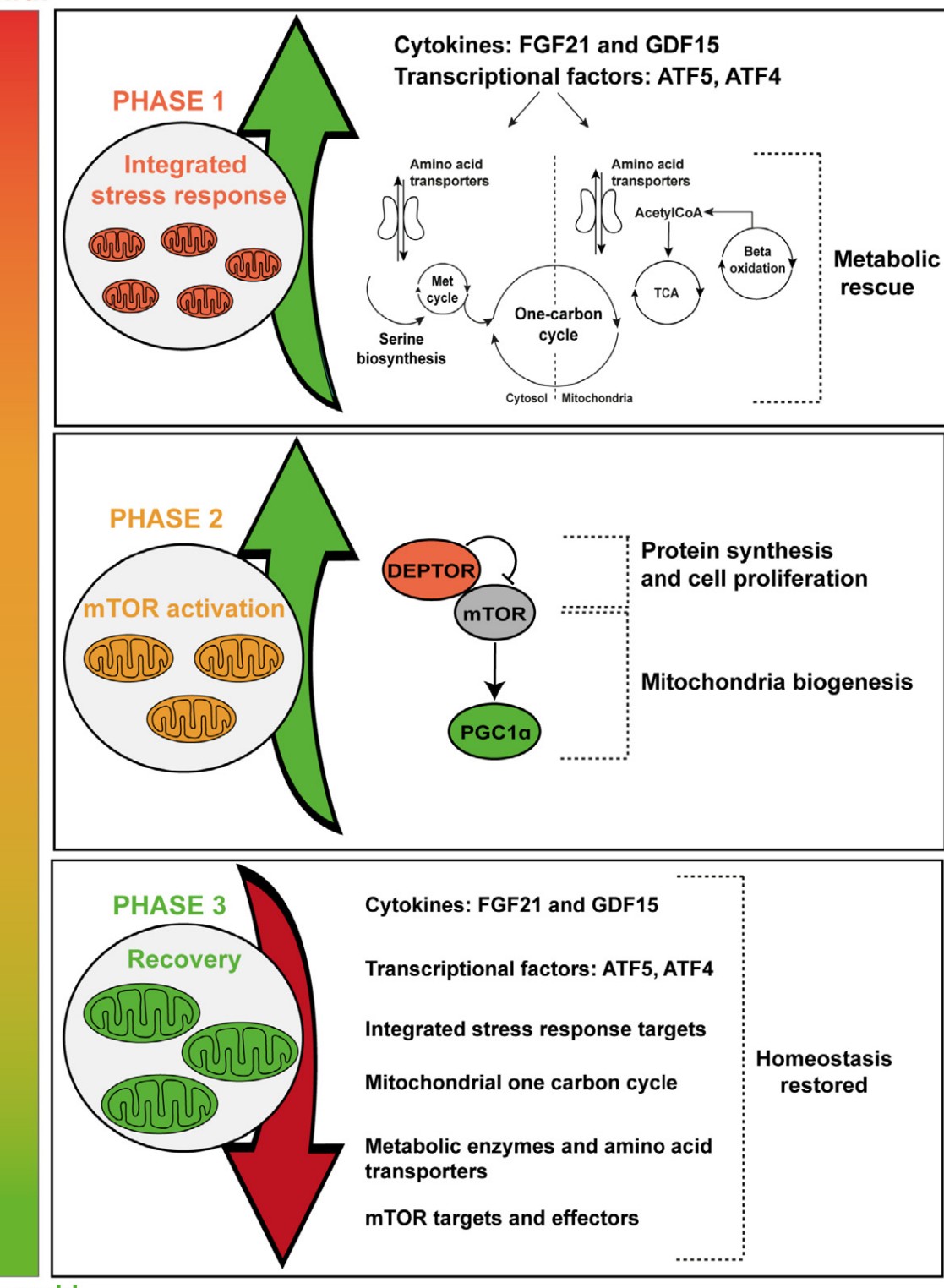

**Figure 6. Graphical summary of changes identified in RIRCD muscle.**

Schematic representation of the metabolic changes in RIRCD. We propose a mechanism that takes place in three phases. Phase 1 consists in *FGF21* increase concomitant with the activation of the integrated stress response which has a protective effect leading to increased expression of proteins involved in amino acid transport and offers a metabolic rescue by increasing lipid oxidation and TCA. The second phase consists of mTOR activation which leads to increased mitochondrial biogenesis and protein synthesis and in phase 3, the recovery, where we have a decrease in all stress markers.

acid status, which seems to be altered in our patients suggested by GCN2 increase (Fig 3D and E).

In phase 2, after the metabolic crisis is surpassed, we have an activation of mTOR in RIRCD muscle supported by decreased levels of DEPTOR, associated with the activation of mTOR and its downstream targets promoting cell survival and growth (Peterson *et al*, 2009) and increased *PGC1α* transcript (Figs 3F and 6). This leads to increased mitochondrial biogenesis and elevation of mt-tRNA$^{Glu}$ steady state, resolving the block in mitochondrial translation in phase 3 (Cunningham *et al*, 2007). The mtDNA copy number in muscle of the RIRCD patients studied in the symptomatic phase was already > 2 times higher than in similar-age infants and showed further 4–40-fold increase after recovery; thus, this increase cannot be explained by a physiological age-related increase but rather represent a compensatory mechanism that contributes to the recovery of these patients (Morten *et al*, 2007).

The mitochondrial replication machinery communicates with cytoplasmic dNTP pools and the *de novo* serine biosynthesis, which was also altered in RIRCD patients, may participate in the metabolic stress response which may offer rescue in primary or secondary mtDNA instability disorders (Nikkanen *et al*, 2016). This is accompanied by increased mitochondrial biogenesis via mTOR activation, further increasing mtDNA copy numbers leading to mt-tRNA$^{Glu}$ increase and improved mitochondrial translation.

The upregulation of TCA in recovering infants further contributes to the metabolic salvage and facilitates the synthesis of amino acids. In support of our findings, a recent study demonstrated that in a mouse model of abnormal mitochondrial translation (*Mrps12$^{ep/ep}$*) there is a similar mitochondrial response that orchestrates the metabolic rescue by increasing TCA together with an increase in cell proliferation and mitochondrial biogenesis (Ferreira *et al*, 2019).

The skeletal muscle is the largest reservoir of amino acids, particularly glutamate and cysteine which are structural components of skeletal muscle, explaining the muscle-specific presentation of RIRCD (Owen *et al*, 2002). Glutamate is also the most abundant amino acid in breast milk, and its concentration increases in the first 4 months of breastfeeding, suggesting the imminent need for this amino acid in infants (Koletzko, 2018). Although we do not have metabolomic data regarding the levels of amino acids in our patients, the significant changes in genes and proteins involved in amino acid metabolism highlight their importance in RIRCD (Fig 6) (Owen *et al*, 2002; Forsstrom *et al*, 2019).

We investigated the effect of amino acid deprivation (cysteine, glutamic acid and glutamine) in RIRCD fibroblasts and detected respiratory chain deficiency (Fig 4). We showed previously that the respiratory chain defect in RIRCD myoblasts can be rescued by cysteine supplementation, by improving cysteine-dependent thiourydilation and detected reduced level of thiourydilation in skeletal muscle, highlighting that amino acids may regulate oxidative phosphorylation (Boczonadi *et al*, 2013). Furthermore, supplementation with cysteine rescued the defect of mitochondrial protein synthesis in fibroblasts carrying mutations in *TRMU, MTO1* or mt-tRNAs (m.3243A>G, m.8344A>G) (Bartsakoulia *et al*, 2016). This might suggest that the cysteine-dependent thiouridylation of the mt-tRNA$^{Glu}$ catalysed by *TRMU* is impaired, supported by the detection of *TRMU* variants in nine of the 27 in RIRCD patients.

The clinical recovery is characteristic for RIRCD; however, the importance of sufficient supply of amino acids is likely to be relevant in other mitochondrial myopathies. We detected similar alterations of metabolic pathways in a clinical trial using bezafibrate in skeletal muscle of patients with m.3243A>G-related mitochondrial myopathy (Steele *et al*, 2020), where the number of complex IV deficient muscle fibres decreased and cardiac function improved, while FGF21 and GDF15 increased in all patients. The changes in treated patients were accompanied by alterations in fatty acid and amino acid metabolism including glutamine and glutamate and the TCA cycle. In this context, the increased amino acid levels could be part of a compensatory response to the primary OXPHOS defect. However, further studies are needed to explore the role of potentially targetable amino acid sensing pathways and their impact in mitochondrial diseases.

In summary, we show that RIRCD is a skeletal muscle-specific disease, where digenic mutations in mitochondrial and nuclear DNA interact with reduced nutritional intake of amino acids and activate a cascade of metabolic events including integrated stress response activation, FGF21 signalling, mTORC1 activation, metabolic shift to TCA and fatty acid oxidation and enhanced mitochondrial biogenesis. This stress response in turn improves the availability of amino acids (glutamate, cysteine) which facilitates the metabolic shift, which in turn improves mt-tRNA$^{Glu}$ steady state and leads to recovery. Similar digenic nuclear-mitochondrial interactions may be relevant in other human mtDNA-related diseases.

# Materials and Methods

## Experimental models and ethics approval

### Ethical approval

Human fibroblasts were obtained from the Biobank of the MRC Neuromuscular Translational Research Centre (London-Cambridge-Newcastle). All the necessary ethical approvals are available in these facilities (REC reference 08/H0906/28 + 5). Human skeletal muscle was obtained by biopsy. A limited number of age-matched control biopsies were obtained from the Essen Biobank (ethical approval 19-9011-BO) from infants and children who were biopsied for the suspicion of a non-mitochondrial muscle disease, but had normal histology. Patients and their family members have given consent to the study "Genotype and phenotype in inherited neurological diseases" study (REC: 13/YH/0310, end date 30/09/2023, IRAS ID: 2042290) which includes all relevant permission for this work. The consent form contains also the consent for any omics analysis.

## Human skeletal muscle

*Vastus lateralis* muscle needed for this study was collected from patients via needle biopsy. Collected samples were frozen in liquid nitrogen and stored at −80°C. Sample codes used for the RNA sequencing are as follows: (i) for the affected vs control comparison F2/1 M, F2/2 F, F3/2 M, F4/1 F, F5/1 M and (ii) for the affected vs recovered comparison F6/1 M. Samples used for proteomics: F2/1 M, F5/1 M, F3/2 M while for Western blotting we have used F2/1 M, F5/1 M, F3/1 M, F3/2 M. For the determination of copy numbers, we have used samples derived from the following patients (i) affected F6/1 M, F3/2 M, F5/1 M and (ii) recovered F3/1 M, F4/1 F. All controls were sex/age-matched. All

details related gender, age and other additional information can be found in Table EV1.

## Cell lines

The human primary fibroblasts were obtained from the Newcastle Biobank from donors via skin biopsy utilising routine protocols. The samples were obtained from the following individuals: F7/7F-healthy carrier, F7/2M-RIRCD, F7/5F-healthy carrier and F7/3F-RIRCD without a clear second mutation. For more details relating the patients, see Table EV1.

HEK293T cells (+/+) and *PDE12* knockout cells (−/−) and −/− cells expressing WT *PDE12*, p.Glu351Ala (catalytic mutant), Δ16 (coding for *PDE12* lacking 16 first aa), Δ23 mutants (coding for *PDE12* lacking 23 first aa) and p.Arg23Trp *PDE12* Cdna and Human 143B osteosarcoma (HOS) were obtained from Michal Minczuk.

## Whole exome sequencing

We performed whole exome sequencing in 34 individuals, homoplasmic for m.14674T>C and bioinformatics analysis, as described previously (Griffin *et al*, 2014). Variants that were annotated as splice-site, frameshift or stop-loss/gain and also non-synonymous variants that were predicted to be damaging, show evolutionary conservation and were present in a known InterPro domain were considered potentially pathogenic and were interrogated for their predicted *in silico* deleteriousness and previous known association with human disease as described (Taylor *et al*, 2014; Richards *et al*, 2015). Variants were filtered for those that were exclusively present in either affected or unaffected individuals. Expected population minor allele frequencies were obtained from the Human Mitochondrial DataBase (http://www.hmtdb.uniba.it/hmdb/) and the Exome Aggregation Consortium (http://exac.broadinstitute.org/). The NIHR BioResource, UK BioBank and 1000 Genomes datasets were queried for m.14674T>C/G mutation carriers.

## Cell culture

Primary human fibroblasts from controls and patients were grown in high glucose Dulbecco's modified Eagle's medium (DMEM, Thermo Fisher Scientific 11965084) supplemented with 2 mM L-glutamine (Thermo Fisher Scientific 35050061) and 10% foetal bovine serum (Thermo Fisher Scientific 26140) at 37°C, in a humidified 5% $CO_2$ atmosphere. Primary fibroblasts were left to grow in normal conditions and then split between normal and cysteine and glutamine depleted medium.

For the cysteine depletion experiment, cells were incubated in cysteine free medium which was purchased from Thermo Fisher Scientific (21013 DMEM, high glucose, no glutamine, no methionine, no cysteine). Cell culture medium was supplemented with Glutamine (Thermo Fisher Scientific 35050061, 2 mM), Sodium Pyruvate (Thermo Fisher Scientific 11360070, 0.11 mg/ml), 5% dialysed FBS (Thermo Fisher Scientific, A3382001) and 0.02 mM of L-cysteine (Sigma-Aldrich C7352) which represents 10% of the standard DMEM concentration, which was enough to support cell growth. Cells were incubated for 12 days in the cysteine depleted media at 37°C, in a humidified 5% $CO_2$ atmosphere.

For the glutamine depletion experiment, cells were grown in standard high glucose DMEM with no glutamine added (Thermo Fisher Scientific 11965084) supplemented with 5% dialysed FBS (Thermo Fisher Scientific, A3382001) for 12 days at 37°C, in a humidified 5% $CO_2$ atmosphere.

Cell lines to study PDE12 were as cultured as described in Pearce *et al* (2017). HEK293 knockout cell line expressing the cDNA coding for p.Arg23Trp was constructed also as described previously by Pearce *et al* (2017). Briefly, HEK293 Flp-In T-Rex was purchased from Invitrogen-R78007 which allows for the generation of stable, doxycycline-inducible expression of transgenes by FLP recombinase-mediated integration. This system was used to generate the PDE12$^{-/-}$ cell lines which inducible expressed PDE12.Strep2.Flag (PDE12.FST2) and the mutants p.Glu351Ala, Δ16, Δ23 mutants and p.Arg23Trp *PDE12* cDNA. To generate the PDE12$^{-/-}$ and PDE12$^{+/-}$ cell lines, HEK293 were transiently transfected to express a pair of CompoZr ZNFs (Sigma-Aldrich), which targeted the *PDE12* gene locus at exon 1. Cells were then electroporated with pZFN1 and pZFN2 using Cell Line Nucleofector (Lonza) and buffer kit V (Lonza VVCA-1003) applying programme A-023. Seventy-two hours after transfection, single cells were sequenced by Sanger sequencing to identify clones harbouring indels in *PDE12*.

PDE12 cells and Human 143B osteosarcoma (HOS) cell lines were cultured in DMEM (Thermo Fisher Scientific 11965084) containing 2 mM Glutamax (Thermo Fisher Scientific 35050061), 10% FBS (Thermo Fisher Scientific 26140), Penicilin/Streptomycin (Gibco 15140122) at 37°C, in a humidified 5% $CO_2$ atmosphere.

## RNA sequencing

For the RNA sequencing, a number of (i) five patient samples and six controls were used for the affected vs control analysis and one muscle sample (in four technical replicates) was used for the affected vs recovered comparison. RNA was extracted with the mirVana miRNA isolation kit according to the manufactures protocol (Thermo Fisher Scientific AM1560). RNA-seq libraries were prepared with Illumina TruSeq Stranded (Illumina MRZH11124) polyA enriched RNA with Ribo-Zero Human (Illumina 20020594) kit and were sequenced on Ilumina HiSeq 2500 Platform according to the paired-end protocol, as previously described (Burns *et al*, 2018). The quality of sequencing reads was checked with FastQC. Raw sequence reads were trimmed to 50 bp with FastX-toolkit v.0.0.14 (http://hannonlab.cshl.edu/fastx_toolkit/index.html) and aligned to complete Human (hg38) reference genomes, using the STAR aligner v.2.5.3a two-pass protocol that is outlined in the GATK documentation (Dobin *et al*, 2013). Number of reads mapped to Ensmbl genes was counted with HTSeq v.0.9.1 (Anders *et al*, 2015), and the differentially expressed genes were identified using the DESeq2 v.1.12.4 package (Love *et al*, 2014).

## Quantification of mitochondrial DNA copy number

The relative mtDNA copy number per cell was quantified by a multiplex Taqman (Bio-rad 4369510) qPCR assay for three affected patients, two recovered and four healthy controls.

This was performed by amplifying *MT-ND1* (mitochondrial encoded gene) and *B2M* (nuclear encoded gene) with a CFX96™ Real-Time PCR Detection System (Bio-Rad) following the protocol

described previously(Bartsakoulia *et al*, 2016). The primers used for template generation of standard curves and the qPCR reaction are as follows B2M: Fw-CACTGAAAAAGATGAGTATGCC, Rv-AACATTCC CTGACAATCCC; MTND1: Fw-AACATTCCCTGACAATCCC, Rv-AAC ATTCCCTGACAATCCC. The copies per microliter of each template were standardised to $1*10^{10}$, and a serial dilution in $1Log_{10}$ dilution steps was amplified along with the DNA negative control on each qPCR plate. This was performed in 20 µl reactions in a 96 well-plate (Bio-Rad 5496), sealed using microplate "B" plate sealers (VWR 391-1293). The reaction mixture was composed of: 5 µl × 5× Taqman (Bio-rad 4369510), 0.4 µM of reverse and forward primers, 25–50 ng of DNA template, 0.2 µl MyTaq HS DNA polymerase (Bio-line BIO-21112) and PCR-grade autoclaved sterile deionised water (to make up 20 µl reaction mixture). The cycling conditions were as follows: (i) initial denaturation at 95°C for 3 min, (ii) 40 cycles of denaturation at 95°C for 10 s, (iii) annealing and extension at 62.5°C for 1 min. The relative mtDNA copy number was calculated using the $\Delta C_t$ data following the equation: CopyNumber = 2 ($2\Delta^{C_t}$ ) where Delta $C_t$ ($\Delta C_t$) equals the sample $C_t$ of the mitochondrial gene (MTND1) subtracted from the sample $C_t$ of the nuclear reference gene (B2M).

## Aminoacylation of mt-tRNA$^{Glu}$ and mt-tRNA$^{Gln}$

The impact of identified variants on the aminoacylation of mt-tRNA$^{Glu}$ was analysed using RNA isolated from fibroblasts grown in standard DMEM, and in fibroblasts grown in MEM with 1% the amino acid concentration present standard DMEM for 48 h. RNA was extracted from sub-confluent fibroblasts using Trizol (Thermo Fisher Scientific 15596026) according to the manufacture's protocol, and the final RNA pellet was dissolved in 10 mM sodium acetate (Sigma-Aldrich S2889-250G), pH 5.0 at 4°C. For the deacylated (dAc) control, the pellet was resuspended in 200 mM Tris–HCl (Applichem Biochemica A3452) at pH 9.5 and incubated at 75°C for 5 min, followed by RNA precipitation and resuspension in 10 mM sodium acetate (Sigma-Aldrich S2889-250G) buffer pH 5. Next, 5 µg of RNA was separated on a 6.5% polyacrylamide gel (Thermo Fisher Scientific NP0321BOX; 19:1 acrylamide:bisacrylamide) containing 8 M urea in 0.1 M sodium acetate pH 5.0 at 4°C and blotted to Hybond N$^+$ membranes (Amersham RPN303B). Following UV-crosslinking, the blots were washed in hybridisation buffer of 7% SDS (Carl Roth CN30.1), 0.25 M sodium phosphate (Sigma-Aldrich 342483-500G) pH 7.6 for 1 h at 65°C. The membrane was subsequently incubated overnight at 65°C in hybridisation solution with $^{32}$P-labelled antisense RNA probes (Hartman Analytic), generated by *in vitro* transcription by T7 RNA polymerase in the presence of $^{32}$P-labelled alpha-UTP (Hartman Analytic), using linearised templates. After hybridisation, the blots were washed six times with 1× SSC (Sigma-Aldrich S6639) for 15 min at 65°C. Bound probes were detected by phosphor imaging on an Amersham Typhoon Scanner.

## Protein lysate preparation and immunoblotting analysis

Muscle samples derived from three patients and three controls (1 month, 17 years old female and 17 years old male) were lysed in 100 µl of lysis buffer (50 mM Tris–HCl- Applichem Biochemica A3452 (pH 7.8) 150 mM NaCl (Merck 1064041000), 1% SDS (Carl

Roth CN30.1) and Complete Mini-Roche 11873580001) using a manual glass grinder. Control and patient fibroblasts were lysed in RIPA buffer (Sigma-Aldrich R0278) with Complete Mini by pipetting. Then, samples were centrifuged for 5 min at 4°C and 5,000 *g*. Protein concentration of the supernatant was determined by BCA assay (Thermo Fisher 23225; according to the manufacturer's protocol).

For immunoblot studies, 10 µg protein was used in each case, loaded on a gradient polyacrylamide gel (NuPage 4–12% Bis-Tris Protein gels Thermo Fisher WG1402BOX, NP0321BOX) and separated for 120 min at 120 V. Following the separation, proteins were transferred to PVDF membrane (Iblot2 Transfer stacks IB23001) using the iBlot2 system (Thermo Fisher IB21001) according to the manufactures protocol. Membranes were blocked with 5% Milk prepared in PBS-T for 2 h followed by four washing steps using PBS (Gibco 18912014) with 0.1% Tween20 (Sigma-Aldrich P7949-100ML; PBS-T). Membranes were incubated with several primary antibodies (Table EV1) at 4 °C (overnight) and then washed in PBS-T thrice. Horseradish peroxidase conjugated secondary goat anti-rabbit antibody (Thermo Fisher Scientific 31460) or goat anti-mouse antibody (Thermo Fisher Scientific 31430) was diluted at 1:25,000 and added to membranes for 1 h. Next, membranes were washed three times in PBS-T for 10 min. By using the enhanced chemiluminescence, horseradish peroxidase substrate (Super-Signal West Pico 34577 and Super-Signal West Femto 34094; Pierce) signals were detected using a UVItech machine.

## Comparative proteomic analysis

### Cell lysis, sample preparation and trypsin digestion

In total, six muscle samples (*vastus lateralis)* derived from three healthy controls (1 month, 17 years old female and 17 years old male) and three patients were processed independently. Approximately 10 slices of 10 µm of muscle were lysed in 50 µl of lysis buffer (50 mM Tris–HCl [Applichem Biochemica A3452] [pH 7.8] 150 mM NaCl, 1% SDS [Carl Roth CN30.1], and Complete Mini-Roche 11873580001) using a manual glass grinder. Then, samples were centrifuged for 5 min at 4°C and 5,000 *g*. Protein concentration of the supernatant was determined by BCA assay (Thermo Fisher 23225; according to the manufacturer's protocol), and cysteines were reduced with 10 mM of DTT (Roche 10708984001) by incubation at 56°C for 30 min. Next, the free thiol groups were alkylated with 30 mM IAA (Sigma-Aldrich I1149-25G) at room temperature (RT) in the dark for 30. Sample digestion and cleanup were performed using filter-aided sample preparation (FASP) as described previously(Roos *et al*, 2019) with some minor changes: 100 µg of protein lysate was diluted 10-fold with freshly prepared 8 M urea/ 100 mM Tris–HCl (Applichem Biochemica A3452; pH 8.5) buffer and placed on PALL microsep centrifugal device(Merck Z648051; 30 KDa cut-off) and centrifuged at 13,500 *g* at RT for 20 min (all the following centrifugation steps were performed under the same conditions). Three washing steps were carried out with 100 µl of 8 M urea (Sigma-Aldrich U1250)/100 mM Tris–HCl (Applichem Biochemica A3452; pH 8.5). For buffer exchange, the device was washed thrice with 100 µl of 50 mM NH$_4$HCO$_3$ (pH 7.8; Sigma-Aldrich S2889-250G). The digestion buffer contains as follows (final volume of 100 µl): trypsin (Promega V5117; 1:25 w/w, protease to substrate), 0.2 M GuHCl (Sigma-Aldrich G3272-500G) and 2 mM

CaCl$_2$ (Sigma-Aldrich C3306) in 50 mM NH$_4$HCO$_3$ (pH 7.8; Sigma-Aldrich S2889-250G), which was added to the concentrated proteins, and the samples were incubated at 37°C for 14 h. Resulting tryptic peptides were recovered by centrifugation with 50 μl of 50 mM NH$_4$HCO$_3$ (Sigma-Aldrich S2889-250G) followed by 50 μl of ultra-pure water. Afterwards, the resulting peptides were acidified (pH < 3 by addition of 10% TFA (v/v) (Biosolve 213141). All digests were quality controlled as described previously.

### LC-MS/MS analysis

Samples were measured using an Ultimate 3000 nano RSLC system coupled to an Orbitrap Fusion Lumos mass spectrometer (both Thermo Scientific). Peptides were preconcentrated on a 100 μm × 2 cm C18 trapping column for 10 min using 0.1% TFA (v/v) at a flow rate of 20 μl/min. Next, the separation of the peptides was performed on a 75 μm × 50 cm C18 main column (both Pepmap, Thermo Scientific 164567) with a 120 min LC gradient ranging from 3 to 35% of 84% ACN (Biosolve 12041), 0.1% FA (v/v) (Biosolve 69141) at a flow rate of 230 nl/min. MS$^1$ spectra were acquired in the Orbitrap from 300 to 1,500 $m/z$ at a resolution of 120,000 using the polysiloxane ion at $m/z$ 445.12003 as lock mass(Olsen *et al*, 2005), with maximum injection times of 50 ms and ACG target was set at $2.0 \times 10^5$ ions. Top fifteen most intense signals were selected for fragmentation by HCD with a collision energy of 30%. MS$^2$ spectra were acquired in the ion trap at a resolution of 120,000, with maximum injection times of 300 ms, a dynamic exclusion of 15 s. The ACG target was set at $2.0 \times 10^3$ for MS$^2$.

### Label-free data analysis

Data analysis of the acquired label-free MS data was performed using the Progenesis LC-MS software from Nonlinear Dynamics (Newcastle upon Tyne, UK). Raw MS data were aligned by Progenesis which automatically selected one of the LC-MS files as reference. After automatic peak picking, only features within retention time and $m/z$ windows from 0–120 min and 300–1,500 $m/z$, with charge states +2, +3 and +4 were considered for peptide statistics and analysis of variance (ANOVA) and MS/MS spectra were exported as peak lists. Peak lists were searched against a concatenated target/decoy version of the human Uniprot database (downloaded on 22.07.2015 containing 20,273 target sequences) using Mascot 2.4 (Matrix Science, Boston, MA, USA), MS-GF$^+$, X!Tandem and MyriMatch with the help of searchGUI 3.2.5(Vaudel *et al*, 2011). Trypsin was selected as enzyme with a maximum of two missed cleavages, carbamidomethylation of Cys was set as fixed and oxidation of Met was selected as variable modification. MS and MS/MS tolerances were set to 10 p.p.m and 0.5 Da, respectively.

To obtain peptide-spectrum match and to maximise the number of identified peptides and proteins at a given quality, we used Peptide-Shaker software 1.4.0 (http://code.google.com/p/peptide-shaker/). Combined search results were filtered at a false discovery rate (FDR) of 1% on the peptide and protein level and exported using the PeptideShaker features that allow direct re-import of the quality-controlled data into Progenesis. Peptide sequences containing oxidised Met were excluded from further analysis. Only proteins that were quantified with unique peptides were exported. For each protein, average of the normalised abundances (obtained from Progenesis) from the analyses was calculated in order to determine the ratios between the patient muscle and control.

Only proteins which were (i) commonly quantified in all the replicates with (ii) unique peptides, (iii) an ANOVA $P$-value of $\leq 0.05$ (Progenesis) and (iv) an average ratio $\leq \log_2 -2.2$ or $\geq \log_2$ 0.98 were considered as up respectively downregulated. AThe Proteomap was generated using the online available tool (https://www.proteomaps.net/). The annotation of these proteomaps is based on the KEGG database platform, each protein is shown by a polygon, and functionally relevant proteins are arranged as neighbours. Additionally, polygon areas represent protein abundances weighted by protein size.

### Quantification and statistical analysis

Data were plotted using GraphPad Prism v.7.0 software (GraphPad Software, USA) or Origin 6.0 (Origin Lab) and Adobe Illustrator Artwork 24.3 (Adobe Systems). The statistical test and method are indicated in the legend of the figures. $P$-values of less than 0.05 or 0.1 (transcriptomics) were considered statistical significant for all experiments.

# Data and code availability

All the data generated or analysed during this study are included in this article or in the supplemental methods and are available from the corresponding author upon request. The exome sequencing data are made available via EGA and RD-Connect (Ucam-horvath dataset) RNA seq data has been deposited to the European Genome-Phenome Archive with the data set identifier EGAS00001004647 (https://ega-archive.org/studies/EGAS00001004647). The mass spectrometry proteomics data have been deposited to the ProteomeXchange Consortium via the PRIDE (Perez-Riverol *et al*, 2019) partner repository with the data set identifier PXD020181 (http://www.ebi.ac.uk/pride/archive/projects/PXD020181).

All code used for the exome sequencing and RNA-seq are publicly available packages that can be found on Bioconductor (https://www.bioconductor.org/) or can be provided on request.

**Expanded View** for this article is available online.

### Acknowledgements

RH was supported by the European Research Council [309548], the Well-come Investigator Award [109915/Z/15/Z]. the Medical Research Council (UK) [MR/N025431/1]; the Wellcome Trust Pathfinder Scheme [201064/Z/16/Z], the Newton Fund [UK/Turkey, MR/N027302/1], the Lily Foundation and the Evelyn Trust. PFC is a Wellcome Trust Principal Research Fellow (212219/Z/18/Z) and a UK NIHR Senior Investigator, who receives support from the Medical Research Council Mitochondrial Biology Unit (MC_UU_00015/9), the Medical Research Council (MRC) International Centre for Genomic Medicine in Neuromuscular Disease, the Evelyn Trust, and the National Institute for Health Research (NIHR) Biomedical Research Centre based at Cambridge University Hospitals NHS Foundation Trust and the University of Cambridge. The views expressed are those of the author(s) and not necessarily those of the NHS, the NIHR or the Department of Health. DH and AR gratefully acknowledge the financial support by the Ministerium für Innovation, Wissenschaft und Forschung des Landes Nordrhein-Westfalen and the Bundesministerium für Bildung und Forschung. This work was also supported by a grant of the French Muscular Dystrophy Association (AFM-Téléthon; #21466) to AR. HL received funding from the Canadian Institute of

Health and Research (CIHR FDN-167281). RDSP is supported by a Medical Research Council Clinician Scientist Fellowship (MR/S002065/1). Part of this work was undertaken in the UCLH/UCL Queen Square Institute of Neurology sequencing facility, which received a proportion of funding from the Department of Health's NIHR BRC funding scheme. The clinical and diagnostic "Rare Mitochondrial Disorders" Service in London is funded by the UK NHS Highly Specialised Commissioners. We thank the NIHR BioResource volunteers for their participation, and gratefully acknowledge NIHR BioResource centres, NHS Trusts and staff for their contribution. We thank the National Institute for Health Research and NHS Blood and Transplant. The views expressed are those of the author(s) and not necessarily those of the NHS, the NIHR or the Department of Health. The analysis of allele frequencies in the UK Biobank Resource was conducted under application number 18794. We thank Reinaldo Issado Takata and Alessandra de la Rocque Ferreira for molecular studies and Dr. Luisa Iommarini (University of Bologna) for the fruitful discussion about the cysteine depletion experiment.

## Author contributions

HG, MJJ, WW, CC and VM performed bioinformatics analysis. HG, MJJ, MaG and EPH were involved in RNA sequencing. DH and AR completed the proteomics studies. DH, MiG, CP, SFP and VB performed cell culture experiments. BM studied mtDNA copy numbers, AP, JD, and BM performed Sanger sequencing. JP, US, ADM, RDSP, MGH, KJ, AC, JFP, MMN, AM, KC, SS, JU, MH, MT, SDM and RH contributed patients to the study. AG-D, VM, HL, MM and PFC contributed constructive comments on bioinformatics analysis and interpretation of data. HG, DH, JSM and RH were involved in the design of the experiments, data analysis and writing of the manuscript.

## Conflict of interest

The authors declare that they have no conflict of interest.

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
