## [Review Process File · The EMBO Journal]

Metabolic shift underlies recovery in reversible infantile respiratory chain deficiency

Denisa Hathazi, Helen Griffin, Matthew Jennings, Michele Giunta, Christopher Powell, Sarah Pierce, Benjamin Munro, Wei Wei, Veronika Boczonadi, Joanna Poulton, Angela Pyle, Claudia Calabrese, Aurora Gomez-Duran, Ulrike Schara, Robert Pitceathly, Michael Hanna, Kairit Joost, Ana Cotta, Julia Paim, Monica Navarro, Jennifer Duff, Andre Mattmann, Kristine Chapman, Serenella Servidei, Adela Della Marina, Johanna Uusimaa, Andreas Roos, Vamsi Mootha, Michio Hirano, Mar Tulinius, Manta Giri, Eric Hoffman, Hanns Lochmüller, Salvatore DiMauro, Michal Minczuk, Patrick Chinnery, Juliane Mueller, and Rita Horvath

DOI: [10.15252/embj.2020105364](https://doi.org/10.15252/embj.2020105364)

Corresponding author(s): Rita Horvath (rh732@medschl.cam.ac.uk)

Review Timeline:

Submission Date:	21st Apr 20
Editorial Decision:	26th May 20
Revision Received:	3rd Jul 20
Editorial Decision:	14th Aug 20
Revision Received:	31st Aug 20
Accepted:	9th Sep 20

Editor: Elisabetta Argenzio

Transaction Report:

Thank you for submitting your manuscript entitled "Metabolic shift underlies recovery in reversible infantile respiratory chain deficiency" [EMBOJ-2020-105364-T] to The EMBO Journal. The study has now been seen by three referees, whose comments are provided below.

As you can see, the referees concur with us on the overall interest of your findings. However, they also raise several critical points that need to be addressed before they can support publication in The EMBO Journal.

Given the overall interest of your study, I am pleased to invite submission of a revised manuscript as indicated in the referee's reports. I would like to point it out that addressing all of the referees' points in a conclusive manner, as well as a strong support from the referees, will be essential for publication in The EMBO Journal.

Please note that it is The EMBO Journal policy to allow only a single major round of revision and it is therefore important to resolve the main concerns at this stage.

When preparing your letter of response to the referees' comments, bear in mind that this will form part of the Review Process File and therefore will be available online to the community. For more details on our Transparent Editorial Process, please visit our website: http://emboj.embopress.org/about#Transparent_Process.

Before submitting your revision, primary datasets (and computer code, where appropriate) produced in this study need to be deposited in an appropriate public database (see <http://msb.embopress.org/authorguide#dataavailability>). Remember to provide a reviewer password if the datasets are not yet public.

We usually expect to receive revised manuscripts within three months of the first decision. We are aware that many laboratories cannot function at full capacity during the current COVID-19/SARS-CoV-2 pandemic and may relax this deadline. Also, we can extend our 'scooping protection policy' to cover the period required for a full revision to address all of the referees' points. Please inform us as soon as a paper with related content published elsewhere.

Thank you again for the opportunity to consider this work for publication, and please feel free to contact me with any questions about submission of the revised manuscript to The EMBO Journal. I look forward to your revision.

Referee #1:

The article of Horvath group is a continuation of their pioneering genetic and mechanistic research on a peculiar mitochondrial disease, previously called reversible COX-deficiency, now more appropriately termed reversible infantile respiratory chain deficiency, RIRCD. Mitochondrial diseases are typically progressive, but this disease, although causing severe muscle weakness in infants, shows spontaneous partial or total recovery around 1 years of age. The penetrance of the trait is incomplete, and according to the authors, only 1:100 of the gene variant carriers manifest a disease. Why so few manifest the severe disease has remained open.

The aim of the genetic study in this article was to identify whether a second gene mutation contributes to severe RIRCD manifestation. The authors succeeded to collect, together with a large international consortium, an exceptional material of 42 subjects with the variant m.14674T>C in mt-tRNA-Glu, which has been previously connected to RIRCD by these authors. The patient material contains disease manifesting subjects and carriers, from 8 different countries, all having the same homoplasmic mtDNA variant. Furthermore, muscle biopsy material was collected, giving an excellent opportunity for molecular pathology studies.

In addition to the mtDNA m.14674T>C variant, their NGS-exome analysis identified in 24 out of 27 affected patients mutations in EARS2 gene, encoding the mitochondrial glutamate tRNA synthetase, which mediates aminoacylation of mt-tRNA-Glu. The few remaining patients had variants in other genes, interestingly also related to glutamate or glutamine metabolism. The data of EARS2 is solid and exciting, and the other genes, even if present in only few patients, offer a surprisingly clear conclusion: nuclear gene variants that affect glutamate metabolism or mt-tRNA-Glu aminoacylation, contribute to manifestation of RIRCD, when co-occurring with the mtDNA m.14674T>C variant. The results indicate that variants in genes affecting mitochondrial metabolism may contribute to mtDNA tRNA-related disease manifestations, which are exceptionally heterogeneous in nature. Therefore, the findings have high relevance beyond RIRCD. Indeed, this is the first evidence of digenic mtDNA-nuclear gene variant contribution to disease.

Major comments:

1) The genetic findings are one of the most solid parts of the paper, but no figures are included in the actual article of these findings; all are in Suppl data files. Table 1 is missing from my article files; it is said to include description of the patients. Please include Table 1 and also shift Suppl table 3 to the actual article file, as it contains essential data of the contributing genes. Provide a figure that shows the mutation sites in the nuclear-encoded proteins (EARS2 and others), the conservation of those sites in species, and provide also in the same figure information in which functional domains the mutations reside. Include a schematic illustration of the mechanisms how the contributing glutamate metabolic genes are proposed to affect manifestation of RIRCD.

2) The authors then utilized their muscle biopsy materials, which they had from some manifesting patients and age-matched controls, and of one recovered patient. Such material is quite exceptional from children. How did the authors get muscle samples from healthy age-matched child controls? This should be described.

3) The muscle samples were subjected to a multiomics approach - proteomics and RNAseq, and the data are of high quality. Their study identifies in the child patients induction of the mitochondrial integrated stress response, ISRmt, previously reported in adult-onset mitochondrial myopathy, in a series of papers from Suomalainen group (Tynismaa et al. 2010, Suomalainen et al. 2011, Nikkanen et al. 2016, Khan et al. 2017) and Larsson group (Kuhl et al. 2017) with upregulation of ATF5, GDF15 and FGF21, serine biosynthesis, activation of mTOR, as well as EIF2alpha phosphorylation. The authors propose a stage-wise activation of the pathway based on their one patient sample after recovery, similar to what was recently found to occur in ISRmt progression in mice by Forsström et al 2019. Even if from one patient, the data are intriguing concerning the recovery, and alleviation of ISRmt along the disease. This suggests that FGF21 and GDF15, biomarkers of mitochondrial diseases, could be used to follow treatment responses and disease progression/recovery. This could be commented in the discussion. Overall, ISRmt components in human muscle and its dynamic progression in mice have been previously described, but not in children, and not to follow recovery. The data are important.

4) The authors find ISRmt in their omics analysis, but then go forward to study ER-stress-related proteins (figure 2). The reason for this is not completely clear to this reviewer. They try to make a point of ER-stress contribution, but the blots of PERK and EIF2alpha phosphorylation, or S6 or mTOR are not convincing. S2448 mTOR shows variability in signals and not convincing changes, and its target T421/S424 S6K as well. VDAC1 signal is not publication quality. No good antibodies against ATFs exists which is why the western analysis of ATF4 is not convincing - to make a clear point, they should replicate it with ATF4 siRNA or study ATF4 nuclear localization by immunofluorescence. ATF5 induction is, however, evident by RNAseq. To this reviewer, the westerns of GCN2 and SLC7A11 are solid - both being part of ISRmt. Taken together, all other signaling westerns except the two, are unconvincing, as is the conclusion of ER stress induction. ISRmt and ISRer are partially overlapping, so ER-stress does not need to be induced even if ISRmt is. Please either improve experimentation of ER-stress and analyze additionally ATF6 induction and XBP1 splicing related to ER-stress. I favor omission of the data; the signature of ISRmt is clear as such.

5) Stress response mechanism, Figure 4. Their own data beautifully show induction of ISRmt, but then in the mechanism figure, the authors emphasize the role of ATF4 and UPRmt, neither of which is backed up by their evidence. The role of ATF4 in muscle has not been verified in mammalian mitochondrial muscle disease - most evidence comes from cultured cells (Quiros et al. 2017). Previous evidence (Forsström et al) and also the current one points to ATF5 contribution (RNAseq) in the muscle. UPRmt response, mostly described in worms, includes induction of heat shock proteins, which are not changed in this articles results. Please revise to reflect the findings of this paper and previous relevant literature.

6) The authors state that de novo serine biosynthesis is induced, but show no data of it.

Minor comments:

7. Figure 2G. MtDNA copy number - always increases during the first year of life (articles by J Poulton). Is the increase related to age or recovery? Include patient ages to the figure.

8. If mtDNA depletion is disease-induced, what is the mechanism? ISRmt has been shown to cause nucleotide precursor changes in muscle of adult patients with mitochondrial myopathy (Nikkanen et al. 2016). Is RIRCD actually a mitochondrial DNA depletion syndrome, or how do the authors explain the mtDNA copy nr decrease?

9. One of the most induced proteins in ISRmt is MTHFD2 of the mitochondrial folate cycle. Induced or not?

Referee #2:

Hathazi et al describe 27 affected individuals with RIRCD and 15 unaffected controls. RIRCD patients carry homoplasmic m.14674T>C mt-tRNAGlu mutations, but only about a third of carriers develop symptoms. Almost all RIRCD patients recover completely from the disease after 6 mo of age. To explain the reduced penetrance of the disease and to define the molecular mechanism of the reversibility of the symptoms, the authors use genomic sequencing and analyse patients' muscle biopsies by transcriptomics and proteomics. WES of 6 RIRCD patients identified heterozygous mutations in various nuclear genes related to mt-tRNAGlu function such as EARS2 and TRMU, indicating that RIRCD is a digenic disease. Transcriptomic and proteomic approaches point to an activation of the integrated stress response pathway (ISR) in response to OXPHOS deficiency, mTORC1 activation and a metabolic shift affecting various pathways including fatty acid

oxidation and amino acid catabolism. These findings are reminiscent of previous observations in patients with mitochondrial dysfunction. As a novel finding, the authors observe reduced ISR in one patient that recovered from RIRCD. Based on these findings, the authors propose a model for the recovery, which includes three phases for recovery: ISR activation, mTORC1 activation and stimulation of fatty acid oxidation and amino acid metabolism, and finally increased mitochondrial biogenesis.

The characterization of RIRCD as a digenic disease is very interesting and could very well explain the limited penetrance of this disease, although as for other mitochondrial diseases the observed tissue specificity remains unexplained. However, insight into the molecular basis for the recovery of some of the patients remains very limited. The evidence provided using OMICS approaches does not establish causality nor does it sufficiently support the in part far-reaching conclusions of the authors. ISR activation is well established as a cellular response to mitochondrial OXPHOS deficiencies and is known to involve mTORC1 activation. Similarly, metabolic shifts in response to impaired OXPHOS have been reported. It therefore remains enigmatic why some RIRCD patients recover. The analysis of one recovered patient shows reduced ISR but whether this indeed contributes to recovery (as speculated by the authors) or reflects the reduced stress response as a result of recovery is unclear. Moreover, the authors propose a three-phase-model for the recovery in some patients, but the analysis of muscle biopsies only provides insight into the steady state situation.

Specific points:

1. The authors list metabolic enzymes that are significantly changed in the proteome or transcriptome in the text but often do not show these data. It would be informative to see the most significant changes in the transcriptomic and proteomic data set in an unbiased manner, rather than showing only selected pathways as in Figure 1C-D.
2. On page 11, the authors state that there is an increase in mTOR signal as well as increased PGC1a and mitochondrial biogenesis. However, the western blot data shows very little significant changes. Are these claims possibly supported by OMICS data?
3. Page 11: The authors refer to increased S2448mTOR in muscle in the text but show in Figure 2C-D that mTOR phosphorylation at S2448 and PGC1a levels are not significantly altered? They also refer to altered levels of DEPTOR, PIK3R3, PIK3CD and PIK3CG but supporting data are not shown.
4. There are a number of mistakes/inconsistencies in the manuscript, concerning data presentation and statistical evaluation of the data. p values are given although only two patients were analysed.

Referee #3:

In this interesting and exciting manuscript, Horvath et al. present data to suggest the mechanism by which infants born with Reversible infantile respiratory chain deficiency (RIRCD) can recover spontaneously after 6 months of age by changes in amino acid availability.

There is no doubt that the authors have put a lot of work and thinking into this study and provide evidence of the metabolic changes in RIRCD. The key strength of the manuscript is the use of genomic sequencing and database to investigate the markedly reduced penetrance of RIRCD. Through their investigation of potential genetic contributors, researchers found additional heterozygous mutations in nuclear genes interacting with mt-tRNAGlu including EARS2 and TRMU in the majority of affected individuals, but not in healthy carriers of m.14674T>C suggesting the additive effects of these variants underlie the phenotype.

The authors demonstrate that the digenic mutations in mitochondrial and nuclear DNA interact

with reduced nutritional intake of amino acids and activate several metabolic events which induce stress response signaling and FGF21 signaling, metabolic shift to TCA and fatty acid oxidation and enhanced mitochondrial biogenesis. Altogether, these pathways increase amino acids availability and enable the metabolic shift.

However, there are several shortcomings that addressing them would strengthen this manuscript. The authors do a great job measuring abundance of mitochondrial proteins and suggest a metabolic switch. However, in order to claim that mitochondrial function has improved, mitochondrial protein abundance is not an adequate measure for overall metabolism. The study would benefit if the authors could provide additional measures of mitochondrial function such as respirometry. As an alternative to testing respiration authors can perform metabolic flux analysis. The authors suggest increase in amino acid metabolism and demonstrate changes in genes and proteins involved in amino acid metabolism however there is no functional data to demonstrate this. Adding any measure that would provide evidence for altered amino acid metabolism will also strengthen the claims of this study.

Specific comments:

In figure 2G authors show that recovery is associated with even further increase in mitochondrial DNA copy number. How can that be explained? Why do the control subjects have reduced mitochondrial DNA compared with both the affected and the control individuals?

In figure 2A-F authors present only a single recovered subject, which is being used to conclude changes in protein expression. Additional recovered subjects would strengthen the claim. With the current hypothesis of the paper the presentation of the recovered subject data is key to the conclusion of the study. In 2G authors present n=2 recovered subjects, why are these two subjects not included in 2A-2F.

In figure 3 authors present data on cysteine and glutamine depletion however it would be interesting to show as a reference data on other amino acids.

Authors describe how they showed previously that the respiratory chain defect in RIRCD myoblasts (Boczonadi et al., 2013) and in fibroblasts carrying mutations in TRMU, MTO1 or mt-tRNAs (Bartsakoulia, Mueller et al., 2016) can be rescued by cysteine supplementation by improving cysteine dependent thiourydilation, so why not perform this rescue experiment with patient fibroblasts? Can this be shown with the digenic mutation found in this study?

Minor Comments:

In the western blot images the authors might want to use abbreviations or small font for some of the titles, in order to better align the sample lanes with their appropriate group.

Graphs in figure 3B and 3D are too small compared to the western blot images

In figure 4B, can the statistical analysis of the strength of this correlation be presented on the graph, and not just in the text?

Figure 5 is beautiful and interesting for people that read the entire paper with enthusiasm, however most readers would look at this figure without reading the entire paper. Modifying then figure so that it is simpler to understand for a person who was browsing the figures would convey the message more clearly and increase the readership.

Point-by-point answers to the referees**Referee #1:**

The article of Horvath group is a continuation of their pioneering genetic and mechanistic research on a peculiar mitochondrial disease, previously called reversible COX-deficiency, now more appropriately termed reversible infantile respiratory chain deficiency, RIRCD. Mitochondrial diseases are typically progressive, but this disease, although causing severe muscle weakness in infants, shows spontaneous partial or total recovery around 1 years of age. The penetrance of the trait is incomplete, and according to the authors, only 1:100 of the gene variant carriers manifest a disease. Why so few manifest the severe disease has remained open.

The aim of the genetic study in this article was to identify whether a second gene mutation contributes to severe RIRCD manifestation. The authors succeeded to collect, together with a large international consortium, an exceptional material of 42 subjects with the variant m.14674T>C in mt-tRNA-Glu, which has been previously connected to RIRCD by these authors. The patient material contains disease manifesting subjects and carriers, from 8 different countries, all having the same homoplasmic mtDNA variant. Furthermore, muscle biopsy material was collected, giving an excellent opportunity for molecular pathology studies.

In addition to the mtDNA m.14674T>C variant, their NGS-exome analysis identified in 24 out of 27 affected patients mutations in EARS2 gene, encoding the mitochondrial glutamate tRNA synthetase, which mediates aminoacylation of mt-tRNA-Glu. The few remaining patients had variants in other genes, interestingly also related to glutamate or glutamine metabolism. The data of EARS2 is solid and exciting, and the other genes, even if present in only few patients, offer a surprisingly clear conclusion: nuclear gene variants that affect glutamate metabolism or mt-tRNA-Glu aminoacylation, contribute to manifestation of RIRCD, when co-occurring with the mtDNA m.14674T>C variant. The results indicate that variants in genes affecting mitochondrial metabolism may contribute to mtDNA tRNA-related disease

manifestations, which are exceptionally heterogeneous in nature. Therefore, the findings have high relevance beyond RIRCD. Indeed, this is the first evidence of digenic mtDNA-nuclear gene variant contribution to disease.

We thank the reviewer for the positive comments.

Major comments:

1) The genetic findings are one of the most solid parts of the paper, but no figures are included in the actual article of these findings; all are in Suppl data files. Table 1 is missing from my article files; it is said to include description of the patients. Please include Table 1 and also shift Suppl table 3 to the actual article file, as it contains essential data of the contributing genes.

We included Table 1 and Supplementary Table 3 (now Table 2) in the actual article file, as the reviewer suggested.

Provide a figure that shows the mutation sites in the nuclear-encoded proteins (EARS2 and others), the conservation of those sites in species, and provide also in the same figure information in which functional domains the mutations reside. Include a schematic illustration of the mechanisms how the contributing glutamate metabolic genes are proposed to affect manifestation of RIRCD.

We thank the reviewer for this suggestion. We included a new figure showing the point mutation sites in the nuclear-encoded proteins (EARS2, TRMU, QRLS1, GOT2 and GLS), the conservation of the amino-acid in species, and information in which functional domains the mutations reside (Figure 1A). We additionally incorporated a schematic diagram showing how these genes interact with mitochondrial translation or mt-tRNA^{Glu} (Figure 1B). We show that the affected proteins contribute to i) mitochondrial protein translation (EARS2 which aminoacylates mt-tRNA^{Glu} and mt-tRNA^{Gln} and QRLS1 which transamidates Glu-tRNA^{Gln} to form the correctly charged Gln-tRNA^{Gln}), ii) amino-acid metabolism (GOT2 which catalyses the interconversion of oxaloacetate and glutamate into aspartate and α -ketoglutarate while GLS metabolizes glutamine to ammonia and glutamate which is further catabolized to the TCA intermediate α -ketoglutarate to fuel the mitochondrial carbon pool) and iii)

COX1 assembly and synthesis due to the dual role of MSS51. MSS51 is not included in Figure 1A as the identified mutation in our patients is a frameshift mutation leading to a premature stop codon and most likely no protein being translated.

2) The authors then utilized their muscle biopsy materials, which they had from some manifesting patients and age-matched controls, and of one recovered patient. Such material is quite exceptional from children. How did the authors get muscle samples from healthy age-matched child controls? This should be described.

“A limited number of age matched control biopsies were obtained from the Essen Biobank (ethical approval 19-9011-BO) from infants and children who were biopsied for the suspicion of a non-mitochondrial muscle disease, but had normal muscle histology.” We added this to the Methods on page 19.

3) The muscle samples were subjected to a multiomics approach - proteomics and RNAseq, and the data are of high quality. Their study identifies in the child patients induction of the mitochondrial integrated stress response, ISRmt, previously reported in adult-onset mitochondrial myopathy, in a series of papers from Suomalainen group (Tyynismaa et al. 2010, Suomalainen et al. 2011, Nikkanen et al. 2016, Khan et al. 2017) and Larsson group (Kuhl et al. 2017) with upregulation of ATF5, GDF15 and FGF21, serine biosynthesis, activation of mTOR, as well as EIF2alpha phosphorylation. The authors propose a stage-wise activation of the pathway based on their one patient sample after recovery, similar to what was recently found to occur in ISRmt progression in mice by Forsström et al 2019. Even if from one patient, the data are intriguing concerning the recovery, and alleviation of ISRmt along the disease. This suggests that FGF21 and GDF15, biomarkers of mitochondrial diseases, could be used to follow treatment responses and disease progression/recovery. This could be commented in the discussion. Overall, ISRmt components in human muscle and its dynamic progression in mice have been previously described, but not in children, and not to follow recovery. The data are important.

We agree with the reviewer and to emphasize the importance of our findings we added a paragraph to the discussion (page 16):

“Our data based on the investigation of a limited number of muscle biopsies suggest that the alleviation of the mitochondrial ISR along the disease is associated with the clinical recovery (**Figure 3F**). In contrast to progressive mitochondrial myopathies that have a long term pathogenic ISR, RIRCD patients present with a milder short term ISR as we observed a significant decrease in related proteins in recovered muscle (**Figure 3E-F**) which induces a beneficial hormetic response by activating protective cellular mechanisms that aid cells against amino-acid deprivation, metabolic insults and oxidative stress (Pakos-Zebrucka *et al.*, 2016). Additionally, this suggests that FGF21 and GDF15, which are biomarkers of mitochondrial diseases, could be used to follow treatment responses and disease progression/recovery in patients with mitochondrial disease.”

4) The authors find ISRmt in their omics analysis, but then go forward to study ER-stress-related proteins (figure 2). The reason for this is not completely clear to this reviewer. They try to make a point of ER-stress contribution, but the blots of PERK and EIF2alpha phosphorylation, or S6 or mTOR are not convincing. S2448 mTOR shows variability in signals and not convincing changes, and its target T421/S424 S6K as well. VDAC1 signal is not publication quality. No good antibodies against ATFs exists which is why the western analysis of ATF4 is not convincing - to make a clear point, they should replicate it with ATF4 siRNA or study ATF4 nuclear localization by immunofluorescence. ATF5 induction is, however, evident by RNAseq. To this reviewer, the westerns of GCN2 and SLC7A11 are solid - both being part of ISRmt. Taken together, all other signaling westerns except the two, are unconvincing, as is the conclusion of ER stress induction. ISRmt and ISRer are partially overlapping, so ER-stress does not need to be induced even if ISRmt is. Please either improve experimentation of ER-stress and analyze additionally ATF6 induction and XBP1 splicing related to ER-stress. I favor omission of the data; the signature of ISRmt is clear as such.

We would like to clarify that our intention was not to study ER stress in our patients` muscle samples. We were exploring the ISR and key targets involved during the disease phase and after recovery in RIRCD muscle biopsies. By studying PERK our

intention was to exclude its possible contribution to increasing ATF4 levels, as PERK activation was described to increase ATF4 synthesis (Kilberg et al. 2009 and Reich et al. 2020). We agree with the reviewer that ER stress is not a main pathway involved in this condition thus we omitted it from our data. We provide evidence that the ISRmt in RIRCD acts also via the GCN2-ATF4 axis, besides the canonical way via ATF5, leading to increased expression of proteins involved in amino-acid transport, lipid oxidation and TCA. We used the ATF4 D4B8 rabbit antibody from Cell Signalling to detect ATF4 via immunoblotting. We believe that this antibody works well as suggested by 252 publications (https://www.citeab.com/antibodies/654138-11815-atf-4-d4b8-rabbit-mab?utm_campaign=Widget+All+Citations&utm_medium=Widget&utm_source=Cell+Signaling+Technology&utm_term=Cell+Signaling+Technology) including some recent papers (Shimizu et al. 2020 (<https://pubmed.ncbi.nlm.nih.gov/32345659/>); Bugallo et al. 2020 (<https://pubmed.ncbi.nlm.nih.gov/32457286/>); Jiang et al. 2020 (<https://pubmed.ncbi.nlm.nih.gov/32439839/>)). As ATF4 is a key factor in linking ISRmt with amino acid metabolism (Quiros et al.2017) we think it may be useful to keep this blot in Figure 3D (previously 2E). We acknowledge that the activation of ATF4 is correlated with its cellular localization as during stress conditions ATF4 migrates to the nucleus where it binds to different DNAs and regulates their transcription (Han et al. 2013). However, immunofluorescence experiments were not possible as we do not have cryosections from the muscles of these patients. Due to the limited amount of muscle material available for our western blots, we had to focus on protein steady state levels of the key factors and we could only perform limited repeats for each blot. We agree with the reviewer and omitted EIF2 α S6K1 mTOR and their phosphorylated forms. Furthermore, we highlighted ISRmt targets from our transcriptomic analysis that are significantly increased in affected patients, similar to the ISRmt proteins described in mice and humans by Forsström et al 2019 (Figure 3F).

5) Stress response mechanism, Figure 4. Their own data beautifully show induction of ISRmt, but then in the mechanism figure, the authors emphasize the role of ATF4 and UPRmt, neither of which is backed up by their evidence. The role of ATF4 in muscle has not been verified in mammalian mitochondrial muscle disease - most evidence comes from cultured cells (Quiros et al. 2017). Previous evidence

(Forsström et al) and also the current one points to ATF5 contribution (RNAseq) in the muscle. UPR^{mt} response, mostly described in worms, includes induction of heat shock proteins, which are not changed in this articles results. Please revise to reflect the findings of this paper and previous relevant literature.

We agree with the reviewer and thank for highlighting the misleading labelling in Figure 6 (previously figure 5). We omitted ER chaperons and UPR^{mt} elements from Figure 6 and from the Discussion. While the ATF family is well conserved as mitochondrial stress sensors and drivers from worms to mammals (Forsström et al 2019), we agree with the reviewer that indeed the role of ATF4 in mammalian mitochondrial muscle diseases is still debated and not fully elucidated. Forsström et al (2019) show that indeed ATF3, 4 and 5 are progressively induced in mouse myoblasts upon decreasing mitochondrial translation and that ATF4 might also be regulated by an ISR^{mt} independent mechanism. It is possible that the regulation of ATF4 during ISR is FGF21 controlled, similar to described in the Deletor mouse (Forsström et al 2019). We cannot exclude a possible contribution of ATF4 in RIRCD. However further studies would be necessary to confirm it.

We added to the Discussion (page 17):

“Our transcriptomic data in agreement with previous evidence points to a clear contribution of *ATF5* to RIRCD ISR^{mt} in human skeletal muscle (Forsstrom *et al.*, 2019). The role of ATF4 in muscle during ISR^{mt} has not been verified in human mitochondrial muscle disease, only in cultured cells (Quiros *et al.*, 2017) and in mouse muscle where ATF4 activation is FGF21 dependent (Forsstrom *et al.*, 2019). While ATF4 has been suggested to be regulated by an ISR^{mt} independent manner in proliferating cells (Forsstrom *et al.*, 2019), we hypothesise that in skeletal muscle of RIRCD patients ATF4 is sensitive to the amino-acid status, which seems to be altered in our patients suggested by GCN2 increase (**Figure 3**).”

6) The authors state that *de novo* serine biosynthesis is induced, but show no data of it.

We thank the reviewer for this comment. We included that the components of the *de novo* serine biosynthesis pathway showed significant change in transcriptomic

analyses in RIRCD muscle in Figure 3, as we detected an increase in enzymes involved in the first 2 steps of *de novo* L-serine synthesis (*PSAT1*, *PHGDH* and *SDSL*).

Minor comments:

7. Figure 2G. MtDNA copy number - always increases during the first year of life (articles by J Poulton). Is the increase related to age or recovery? Include patient ages to the figure.

We are aware of the publication of Jo Poulton`s group demonstrating an age-related increase of liver and muscle mtDNA copy numbers in infants (Morten et al., 2007).

We added the following sentence to the Discussion on page 17.

“The mtDNA copy number in muscle of the RIRCD patients studied in the symptomatic phase was already >2 times higher than in similar-age infants and showed further 4-40-fold increase after recovery, thus this increase cannot be explained by a physiological age-related increase but rather represents a compensatory mechanism that contributes to the recovery of these patients (Morten *et al*, 2007).”

As requested by the reviewer we added the ages of the patients to Figure 3C (previously 2G) as well as the ages of the 4 control samples (3 months, 3, years 4 years and 7 years) to the legend of Figure 3C.

8. If mtDNA depletion is disease-induced, what is the mechanism? *ISRmt* has been shown to cause nucleotide precursor changes in muscle of adult patients with mitochondrial myopathy (Nikkanen et al. 2016). Is RIRCD actually a mitochondrial DNA depletion syndrome, or how do the authors explain the mtDNA copy nr decrease?

We appreciate the suggestion of the reviewer by referring to the paper by Nikkanen et al. 2016. We detected in part similar changes in RIRCD infants, however we

emphasize that muscle mtDNA copy number in RIRCD infants is not low, it is rather higher (>2 times) than in age matched control muscle (as the increase in mtDNA is compensating for the low steady state level of mt-tRNA^{Glu}), therefore RIRCD is not a mtDNA depletion syndrome.

We reference the Nikkanen paper at page 17 in the Discussion:

“The mitochondrial replication machinery communicates with cytoplasmic dNTP pools and the *de novo* serine biosynthesis, which are altered in RIRCD patients and may participate in the metabolic stress response which may offer rescue in primary or secondary mtDNA instability disorders (Nikkanen et al. 2016). This is accompanied by increased mitochondrial biogenesis via mTOR activation (increased mtDNA copy numbers) thus leading to mt-tRNA^{Glu} increase and improved mitochondrial translation.”

9. One of the most induced proteins in ISRmt is MTHFD2 of the mitochondrial folate cycle. Induced or not?

Previously a possible uncoupling of the regulation of mitochondrial folate cycle during ISRmt was suggested (Forsstrom *et al.*, 2019). *MTHFD2* transcript is slightly increased (not statistically significant) in skeletal muscle of the RIRCD patients which was shown to be FGF21 independent. In contrast, the FGF21 controlled *MTHFD1L* shows a significant increase in RIRCD muscle (Figure 3F).

Referee #2:

Hathazi et al describe 27 affected individuals with RIRCD and 15 unaffected controls. RIRCD patients carry homoplasmic m.14674T>C mt-tRNAGlu mutations, but only about a third of carriers develop symptoms. Almost all RIRCD patients recover completely from the disease after 6 mo of age. To explain the reduced penetrance of the disease and to define the molecular mechanism of the reversibility of the symptoms, the authors use genomic sequencing and analyse patients' muscle biopsies by transcriptomics and proteomics. WES of 6 RIRCD patients identified

heterozygous mutations in various nuclear genes related to mt-tRNAGlu function such as EARS2 and TRMU, indicating that RIRCD is a digenic disease. Transcriptomic and proteomic approaches point to an activation of the integrated stress response pathway (ISR) in response to OXPHOS deficiency, mTORC1 activation and a metabolic shift affecting various pathways including fatty acid oxidation and amino acid catabolism. These findings are reminiscent of previous observations in patients with mitochondrial dysfunction. As a novel finding, the authors observe reduced ISR in one patient that recovered from RIRCD. Based on these findings, the authors propose a model for the recovery, which includes three phases for recovery: ISR activation, mTORC1 activation and stimulation of fatty acid oxidation and amino acid metabolism, and finally increased mitochondrial biogenesis.

The characterization of RIRCD as a digenic disease is very interesting and could very well explain the limited penetrance of this disease, although as for other mitochondrial diseases the observed tissue specificity remains unexplained. However, insight into the molecular basis for the recovery of some of the patients remains very limited. The evidence provided using OMICS approaches does not establish causality nor does it sufficiently support the in part far-reaching conclusions of the authors.

We agree with the reviewer that our data do not fully explain the recovery, however we performed a multi-omics analysis from the limited amount of muscle tissue which was available from RIRCD infants. We agree that further experiments with targeting the key components in cell lines or other model systems would be valuable. Unfortunately, we did not have access to muscle cells, and there are no animal models of this condition. We studied fibroblasts, however these do not manifest the molecular defect without additional stressors (such as amino acid deprivation), making any of these targeted analyses very difficult.

While ISR^{mt} and its dynamic progression has been previously described in mice (Forsström et al 2019), this is the first report describing elements of ISR^{mt} in children, and following recovery. Despite having follow up data before and after recovery only from one patient, we show an alleviation of ISR^{mt} during the course of

the disease which gives us valuable biomarkers such as FGF21 and GDF15 that could be used to follow up treatment response and overall disease progression in patients with mitochondrial myopathies.

ISR activation is well established as a cellular response to mitochondrial OXPHOS deficiencies and is known to involve mTORC1 activation. Similarly, metabolic shifts in response to impaired OXPHOS have been reported. It therefore remains enigmatic why some RIRCD patients recover. The analysis of one recovered patient shows reduced ISR but whether this indeed contributes to recovery (as speculated by the authors) or reflects the reduced stress response as a result of recovery is unclear. Moreover, the authors propose a three-phase-model for the recovery in some patients, but the analysis of muscle biopsies only provides insight into the steady state situation.

We highlight that we performed transcriptomic and proteomics studies on 6 affected and 2 recovered patients, however a direct comparison was only available in one patient, where we collected muscle in the symptomatic phase and after recovery from the same individual.

We agree with the reviewer that it is unclear, why similar alterations rescue the disease in RIRCD and not in other mitochondrial diseases such as m.3243A>G and how ISR can be protective (reversible mitochondrial disease) or deleterious (progressive mitochondrial disease). We hypothesize that it is potentially depending on its duration (short term-beneficial by activating specific rescue mechanisms vs long term-pathogenic, leads to cell apoptosis). Future studies performed on suitable model systems, such as patient derived myoblasts/myotubes may be able to provide an answer for this question.

We added to the Discussion on page 18.

“We detected similar alterations of metabolic pathways in a clinical trial using bezafibrate in skeletal muscle of patients with m.3243A>G-related mitochondrial myopathy (Steele et al., 2020), where the number of complex IV deficient muscle fibres decreased and cardiac function improved, while FGF21 and GDF15 increased in all patients. The changes in treated patients were accompanied by alterations in

fatty acid and amino-acid metabolism including glutamine and glutamate and the TCA cycle. In this context the increased amino acid levels could be part of a compensatory response to the primary OXPHOS defect.”

Specific points:

1. The authors list metabolic enzymes that are significantly changed in the proteome or transcriptome in the text but often do not show these data. It would be informative to see the most significant changes in the transcriptomic and proteomic data set in an unbiased manner, rather than showing only selected pathways as in Figure 1C-D.

As the number of genes/proteins showing significant changes is very high we only included some key factors in Figure 2 (previously figure 1) in the original submission. However, we agree with the reviewer that it is important to show changes of metabolic enzymes in an unbiased manner and added Figure EV4, showing a heatmap of the most significantly changed genes. Furthermore, we highlighted ISRmt targets from our transcriptomic analysis that are significantly increased in affected patients, similar to the ISRmt proteins described in mice by Forsström et al 2019 (Figure 3).

2. On page 11, the authors state that there is an increase in mTOR signal as well as increased PGC1a and mitochondrial biogenesis. However, the western blot data shows very little significant changes. Are these claims possibly supported by OMICS data?

We agree with the reviewer that some of the changes seen in our western blotting analysis may not reach statistical significance, therefore we omitted some of these blots from Figure 3. Our OMICS data did not show any alterations in the levels of *mTOR* transcripts however *DEPTOR* transcript, a component of the mTORC1 and mTORC2 complexes shows a significant decrease in affected patients compared to controls, and increases with recovery (Figure 3). The decrease of *DEPTOR* was associated with the activation of mTOR and its downstream targets, promoting cell survival and growth (Peterson et al, 2009). Our transcriptomic data further suggests an activation of mTOR and its downstream targets such as *PGC1 α* (mitochondria biogenesis), *RRAGC* (cellular response to amino-acid availability), *EIF4EBP1*

(protein translation) and *PPARG* (lipid synthesis) (Laplante and Sabatini 2009) in RIRCD compared to controls (Figure 3), while in recovered patients these transcripts decrease.

3. Page 11: The authors refer to increased S2448mTOR in muscle in the text but show in Figure 2C-D that mTOR phosphorylation at S2448 and PGC1 α levels are not significantly altered? They also refer to altered levels of DEPTOR, PIK3R3, PIK3CD and PIK3CG but supporting data are not shown.

We thank the reviewer for the suggestions. As the levels of the mTOR and S2448 mTOR are not statistically significant we omitted these data from our manuscript. As discussed in point 2 above, we provided further evidence from our OMICS data that suggests the activation of mTOR.

We acknowledge that further studies that focus on the phosphorylation of these proteins or translocation to the mitochondria (for PGC1 α activation) would be needed to clearly establish the activation of mTOR and its consequences in RIRCD recovery, however these are not possible due to the limited amount of muscle tissue.

We have included PIK altered proteins in Figure 2A (previously Figure 1A).

4. There are a number of mistakes/inconsistencies in the manuscript, concerning data presentation and statistical evaluation of the data. p values are given although only two patients were analysed.

Concerning RNA sequencing, The DESeq2 method used is able to calculate p values with low replicate (patient) numbers at the expense of statistical power, as described by Sullivan and Feinn (<https://www.ncbi.nlm.nih.gov/pmc/articles/PMC4878611/>). Likewise, other tests such as Student's T test can also be performed with n=2 at the cost of statistical power. Regarding Student's T test there is a limitation in determination of normality of the distribution, but this is true also at n=3 and even at much higher patient numbers.

In our paper the only instance of a 2 patient comparison made for which p values are stated is as a threshold for labelling in Figure 5A, which is a comparison of patients with versus without mutations in *EARS2*. Other uses of this comparison are to show

a trend in dysregulated transcripts of the mitochondrial OXPHOS system and to show a trend of decreasing transcript levels of mtDNA-encoded genes with increasing Gl/Gln residues, for which p values of individual genes are not used (Figure 5B). Transcriptomic studies compare 5 patients to 6 controls, 1 patient before and after recovery (p values are given for technical replication), and proteomic studies compare 5 patients to 3 controls.

In figure 5 although we analysed only 2 patients, 2 carriers and 2 control fibroblasts we would like to underline that the result of the quantification are resulted from three independent experiments thus aiding with the statistical power.

Referee #3:

In this interesting and exciting manuscript, Horvath et al. present data to suggest the mechanism by which infants born with Reversible infantile respiratory chain deficiency (RIRCD) can recover spontaneously after 6 months of age by changes in amino acid availability. There is no doubt that the authors have put a lot of work and thinking into this study and provide evidence of the metabolic changes in RIRCD. The key strength of the manuscript is the use of genomic sequencing and database to investigate the markedly reduced penetrance of RIRCD. Through their investigation of potential genetic contributors, researchers found additional heterozygous mutations in nuclear genes interacting with mt-tRNA^{Glu} including EARS2 and TRMU in the majority of affected individuals, but not in healthy carriers of m.14674T>C suggesting the additive effects of these variants underlie the phenotype. The authors demonstrate that the digenic mutations in mitochondrial and nuclear DNA interact with reduced nutritional intake of amino acids and activate several metabolic events which induce stress response signaling and FGF21 signaling, metabolic shift to TCA and fatty acid oxidation and enhanced mitochondrial biogenesis. Altogether, these pathways increase amino acids availability and enable the metabolic shift. However, there are several shortcomings that addressing them would strengthen this manuscript. The authors do a great job measuring abundance of mitochondrial

proteins and suggest a metabolic switch. However, in order to claim that mitochondrial function has improved, mitochondrial protein abundance is not an adequate measure for overall metabolism. The study would benefit if the authors could provide additional measures of mitochondrial function such as respirometry. As an alternative to testing respiration authors can perform metabolic flux analysis. The authors suggest increase in amino acid metabolism and demonstrate changes in genes and proteins involved in amino acid metabolism however there is no functional data to demonstrate this. Adding any measure that would provide evidence for altered amino acid metabolism will also strengthen the claims of this study.

We agree with the reviewer completely that measuring respiration, metabolic flux and amino acid concentrations would be additional useful methods that would support our findings. Unfortunately, we did not have the possibility to perform these studies in the limited amount of muscle available from these infants and fibroblasts do not manifest the molecular defect without additional triggering factors thus making flux analysis and metabolomics unreliable. Muscle cells (myoblasts, myotubes) may have been a better cell type to perform these experiments, but we did not have access to patients` myoblasts.

Specific comments:

In figure 2G authors show that recovery is associated with even further increase in mitochondrial DNA copy number. How can that be explained? Why do the control subjects have reduced mitochondrial DNA compared with both the affected and the control individuals?

It is correct that the mtDNA copy number in muscle of the RIRCD patients in the symptomatic phase was >2 times higher than in similar-age infants – already illustrating some compensation for abnormal mitochondrial translation, and this showed further significant 4-40-fold increase in parallel with recovery. We think that this is triggered by increased mitochondrial biogenesis as part of the rescue mechanism and leads to increase in mt-tRNA^{Glu} steady state and clinical recovery of these infants.

In figure 2A-F authors present only a single recovered subject, which is being used

to conclude changes in protein expression. Additional recovered subjects would strengthen the claim. With the current hypothesis of the paper the presentation of the recovered subject data is key to the conclusion of the study. In 2G authors present n=2 recovered subjects, why are these two subjects not included in 2A-2F.

We highlight that we performed transcriptomic and proteomics studies on 6 affected and 2 recovered patients, however a direct comparison was only available in one patient from which we collected muscle in the symptomatic phase and after recovery.

In figure 3 authors present data on cysteine and glutamine depletion however it would be interesting to show as a reference data on other amino acids.

We agree with the reviewer completely. In fact, we are planning to perform such experiments with depleting and supplementing different amino acids, however demonstrating that depletion of the most important amino acids in regard of this specific defect (cysteine, glutamine and glutamic acid) triggers the mitochondrial defect in fibroblasts is already supporting our hypothesis. Comprehensive studies sequentially depleting different amino-acids may be also more relevant in myoblasts as amino-acids have different roles in individual tissues (e.g. increased intramuscular glutamine is needed in muscle protein synthesis (Mittendorfer et al. 2001).

Authors describe how they showed previously that the respiratory chain defect in RIRCD myoblasts (Boczonadi et al., 2013) and in fibroblasts carrying mutations in TRMU, MTO1 or mt-tRNAs (Bartsakoulia, Mueller et al., 2016) can be rescued by cysteine supplementation by improving cysteine dependent thiourydilation, so why not perform this rescue experiment with patient fibroblasts? Can this be shown with the digenic mutation found in this study?

We would have liked to perform such studies in RIRCD fibroblasts, however the mitochondrial respiratory chain defect is not expressed in these cells without triggering it by amino acid depletion. Therefore, we did not have a readout for demonstrating rescue after amino acid supplementation.

Minor Comments:

In the western blot images the authors might want to use abbreviations or small font for some of the titles, in order to better align the sample lanes with their appropriate group.

We agree with the reviewer and adjusted font sizes.

Graphs in figure 3B and 3D are too small compared to the western blot images.

We increased the size of the graphs in Figure 4B and 4D (previously 3B and 3D).

In figure 4B, can the statistical analysis of the strength of this correlation be presented on the graph, and not just in the text?

We added p values demonstrating the strength of the correlation to Figure 5 (previously Figure 4).

Figure 5 is beautiful and interesting for people that read the entire paper with enthusiasm, however most readers would look at this figure without reading the entire paper. Modifying then figure so that it is simpler to understand for a person who was browsing the figures would convey the message more clearly and increase the readership.

We thank the reviewer for this suggestion. We simplified the text in Figure 6 (previously Figure 5) and we hope that this version is easier to understand for the readers.

Thank you for submitting your revised manuscript. Please accept my apologies for the delay in getting back to you with our decision due to summer holidays. The study has been re-reviewed by two of the original referees, whose comments are shown below.

As you can see, while the referees find that their criticisms have been sufficiently addressed, reviewer #2 points out that the pathophysiological relevance of the observed alterations remains unclear and invites you to tone down the abstract. In addition, there are a few editorial issues concerning the text and the figures that I need you to address before we can officially accept your manuscript.

Referee #1:

The authors have satisfactorily responded to my criticism. I have no further comments.

Referee #2:

The authors have carefully considered the points of criticisms when revising their manuscript and significantly improved the presentation and discussion of the data. The manuscript establishes beautifully RIRCD as a digenic disease and describes comprehensively associated metabolic changes. However, the limited availability of patient material does not allow to establish causality or a longitudinal analysis of metabolic changes during the recovery phase. I am still concerned that in some cases only one patient has been analysed. Therefore, it remains rather speculative which of the observed alterations are of pathophysiological relevance and whether different recovery phases can indeed be distinguished. The authors discuss their model and its limitations now more convincingly in the discussion but a more careful wording in the abstract still appears appropriate.

We wish to thank you for having our revised manuscript entitled " **Metabolic shift underlies recovery in reversible infantile respiratory chain deficiency**" reviewed by two reviewers. We have addressed the comment made by reviewer 2 and corrected the issues pointed out by the editorial team and would now like to resubmit the revised manuscript. The changes are marked with yellow in the manuscript.

2nd Revision - Editorial Decision**9th Sep 2020**

I am pleased to inform you that your manuscript has been accepted for publication in The EMBO Journal.

Congratulations!

Corresponding Author Name: Dr. Rita Horvath
 Journal Submitted to: The EMBO Journal
 Manuscript Number: EMBOJ-2020-105364-T